# FcγRIIb differentially regulates pre-immune and germinal center B cell tolerance in mouse and human

Marion Espéli[1,2], Rachael Bashford-Rogers[1,3], John M. Sowerby[1,4], Nagham Alouche[2], Limy Wong[1], Alice E. Denton[1,5], Michelle A. Linterman [1,5] & Kenneth G.C. Smith [1,4]

Several tolerance checkpoints exist throughout B cell development to control autoreactive B cells and prevent the generation of pathogenic autoantibodies. FcγRIIb is an Fc receptor that inhibits B cell activation and, if defective, is associated with autoimmune disease, yet its impact on specific B cell tolerance checkpoints is unknown. Here we show that reduced expression of FcγRIIb enhances the deletion and anergy of autoreactive immature B cells, but in contrast promotes autoreactive B cell expansion in the germinal center and serum auto-antibody production, even in response to exogenous, non-self antigens. Our data thus show that FcγRIIb has opposing effects on pre-immune and post-immune tolerance checkpoints, and suggest that B cell tolerance requires the control of bystander germinal center B cells with low or no affinity for the immunizing antigen.

[1] The Department of Medicine, University of Cambridge School of Clinical Medicine, Cambridge Biomedical Campus, Cambridge, CB2 OXY England, UK. [2] UMR996 - Inflammation, Chemokines and Immunopathology, Inserm, Univ Paris-Sud, Université Paris-Saclay, Clamart F-92140, France. [3] Present address: Wellcome Centre for Human Genetics, Roosevelt Drive, Oxford OX3 7BN, UK. [4] Present address: Cambridge Institute of Therapeutic Immunology & Infectious Disease, Jeffrey Cheah Biomedical Centre Cambridge Biomedical Campus, University of Cambridge, CB2 0AW Cambridge, UK. [5] Present address: Lymphocyte Signalling and Development, Babraham Institute, CB22 3AT Cambridge, UK. Correspondence and requests for materials should be addressed to M.E. (email: marion.espeli@inserm.fr) or to K.G.C.S. (email: kgcs2@cam.ac.uk)

Three major tolerance checkpoints during B-cell development control the generation of autoreactive B-cell clones. The first two checkpoints shape the pre-immune repertoire, while the third occurs following antigen-mediated activation[1]. First, the self-reactivity of immature B cells in the bone marrow (BM) is tested, with a large fraction of autoreactive clones being eliminated (a process called central tolerance)[2–4]. Autoreactive B cells that manage to leave the BM are subjected to a second checkpoint as they transit from immature to mature B cells in the blood and spleen, leading to a further decrease in the frequency of autoreactive clones among mature B cells[5,6]. Finally, following B-cell activation, autoreactive B cells may also be eliminated from the germinal center or corrected by somatic hypermutation, further reducing the chance of generating potentially harmful autoreactive memory B cells and autoantibody-producing plasma cells. The last two checkpoints, taking place outside the BM, contribute to peripheral tolerance[1,6,7].

At a cellular and molecular level, several mechanisms of B-cell tolerance have been described. Clonal deletion and anergy were first proposed as major mechanisms regulating B-cell tolerance[2,3,5,8–10]. Receptor editing has also been shown to play a key role in central tolerance in the BM[4,11,12]. Autoreactive B cells can also be excluded from the B-cell follicles, with a lack of T cell help precipitating their subsequent apoptosis[13,14]. The nature and location of specific autoantigens seem to determine which tolerance mechanism is used, with deletion being preferentially induced by high-avidity antigens while low-avidity interactions tend to drive receptor editing and anergy[9–11]. Germinal center (GC) tolerance mechanisms are still only partially characterised. It has been suggested that autoantigens need to be present within the germinal center to properly control autoreactive clones, either by apoptosis[15,16] or by redemption of their B-cell receptor (BCR) via corrective somatic hypermutation[7,17,18].

The strength of BCR signalling is thought to be important in all of these mechanisms, suggesting that inhibitory receptors regulating the BCR threshold of activation may be central actors in B-cell tolerance. The inhibitory receptor FcγRIIb negatively regulates signalling induced by the BCR[19,20] and is involved in the control of autoimmunity; however, its role in the regulation of specific B-cell tolerance checkpoints has not been determined.

This question is important, as reduced FcγRIIB expression or function has been associated with autoimmune disease in both humans and mice. In humans, a single-nucleotide polymorphism in *FCGR2B* leading to the replacement of an isoleucine by a threonine at position 232 (T232I) results in reduced inhibitory function[21,22], and has been associated with susceptibility to SLE[23–27], but protection against malaria[26,28]. Humanised mice reconstituted with cord blood cells bearing the 232T polymorphism display defective B-cell development and produce autoantibodies[29]. Naturally occurring variations have also been described in the promoter of human *FCGR2B*, and it has been suggested these too may predispose to SLE[30,31]. FcγRIIb appears to play a remarkably similar role in mouse and human biology. Thus, FcγRIIb-deficient mice present an exacerbated immune response[32,33], are prone to inducible and spontaneous autoimmunity[34–38] with more autoreactive GC B-cell clones[39], and are protected from malaria[26]. Transgenic mice overexpressing FcγRIIb on B cells display a reverse phenotype, with reduced immune responses and resistance to autoimmune disease[40]. Interestingly, natural variants common in wild mice from the *Mus musculus* genus have also been reported in the promoter region of *Fcgr2b*[41], and have been shown to be associated with autoimmunity-prone laboratory strains[42,43]. By generating a knock-in (KI) mouse model bearing these wild promoter variants on the C57BL/6 background (FcγRIIb^{wild/H1} KI), we

demonstrated that they control the upregulation of FcγRIIb on B cells upon activation. In the absence of FcγRIIb upregulation on GC B cells, we observed an increased number of total GC B cells following T-dependent immunisation, but, surprisingly, the number of those specific for the immunising antigen was normal[44]. This suggested that FcγRIIb upregulation may be necessary to prevent bystander B cells, that is those with no or negligible affinity for the immunising antigen, from persisting or expanding in the GC. Moreover, we observed a transient production of autoantibodies following exogenous antigenic challenge in our FcγRIIb^{wild/H1} KI mice, demonstrating that tight regulation of FcγRIIb expression is necessary to maintain B-cell tolerance[44].

Here, we use an experimental system enabling us to monitor the frequency of autoreactive (or potentially autoreactive) B cells throughout B-cell differentiation and activation, in the context of different levels of FcγRIIb expression and autoantigen exposure. Our results show that a single inhibitory receptor can have contrasting effects on different tolerance checkpoints, limiting pre-immune while promoting post-immune tolerance. Our results suggest a novel mechanism of GC tolerance, in which FcγRIIb prevents the emergence of bystander autoreactive B cells. Moreover, they also suggest that, in this context, GC tolerance is dominant over pre-immune checkpoints for controlling the development of autoimmunity.

## Results

**Reduced FcγRIIb expression increases central tolerance.** Absence or reduced expression of FcγRIIb on B cells has been shown to drive autoimmunity. We thus set out to determine its impact upon the different checkpoints of B-cell tolerance. We used mouse strains expressing reduced, normal or increased levels of FcγRIIb (FcγRIIb-deficient mice (KO), FcγRIIb WT mice (WT) and FcγRIIb-BFcR transgenic mice (BTG) that overexpress FcγRIIb on B cells only[40]), crossed to the SW_{HEL} mouse strain. In SW_{HEL} mice, a transgene encoding an Ig heavy chain specific for the protein Hen Egg Lysozyme (HEL) has been knocked-in to the endogenous heavy chain locus, allowing class-switching and somatic hypermutation. These features, together with the fact that SW_{HEL} mice can generate a polyclonal immune response against other antigens as only 10–30% of their B cells are specific for HEL, enables the physiological tracking of a GC B-cell response following immunisation[45].

We first evaluated the role of FcγRIIb in central tolerance against a membrane-bound autoantigen using the membrane-HEL (mHEL) mouse strain[5]. Lethally irradiated wild-type or mHEL mice were reconstituted with bone marrow from SW_{HEL}-FcγRIIb KO, SW_{HEL}-FcγRIIb WT or SW_{HEL}-FcγRIIb BTG donor mice (Fig. 1a). As previously described[45], the frequency and absolute number of HEL-specific BM immature B cells was significantly reduced in mHEL compared with wild-type recipients irrespective of the donor used, consistent with clonal deletion (Fig. 1b and Supplementary Fig. 1a). Interestingly, deletion was enhanced in mHEL mice reconstituted with SW_{HEL}-FcγRIIb KO BM cells and reduced in mHEL mice reconstituted with SW_{HEL}-FcγRIIb BTG BM cells compared to WT controls (Fig. 1c). Thus, reduced FcγRIIb expression increases the deletion of autoreactive clones at the central tolerance checkpoint.

**Reduced FcγRIIb expression enhances peripheral tolerance.** After leaving the BM, immature B cells experience another round of selection at the transitional B-cell stage[6]. We thus determined whether FcγRIIb expression had an effect on the frequency and number of splenic HEL-specific immature and mature B cells (Fig. 2a and Supplementary Fig. 1b, c). Absence of FcγRIIb

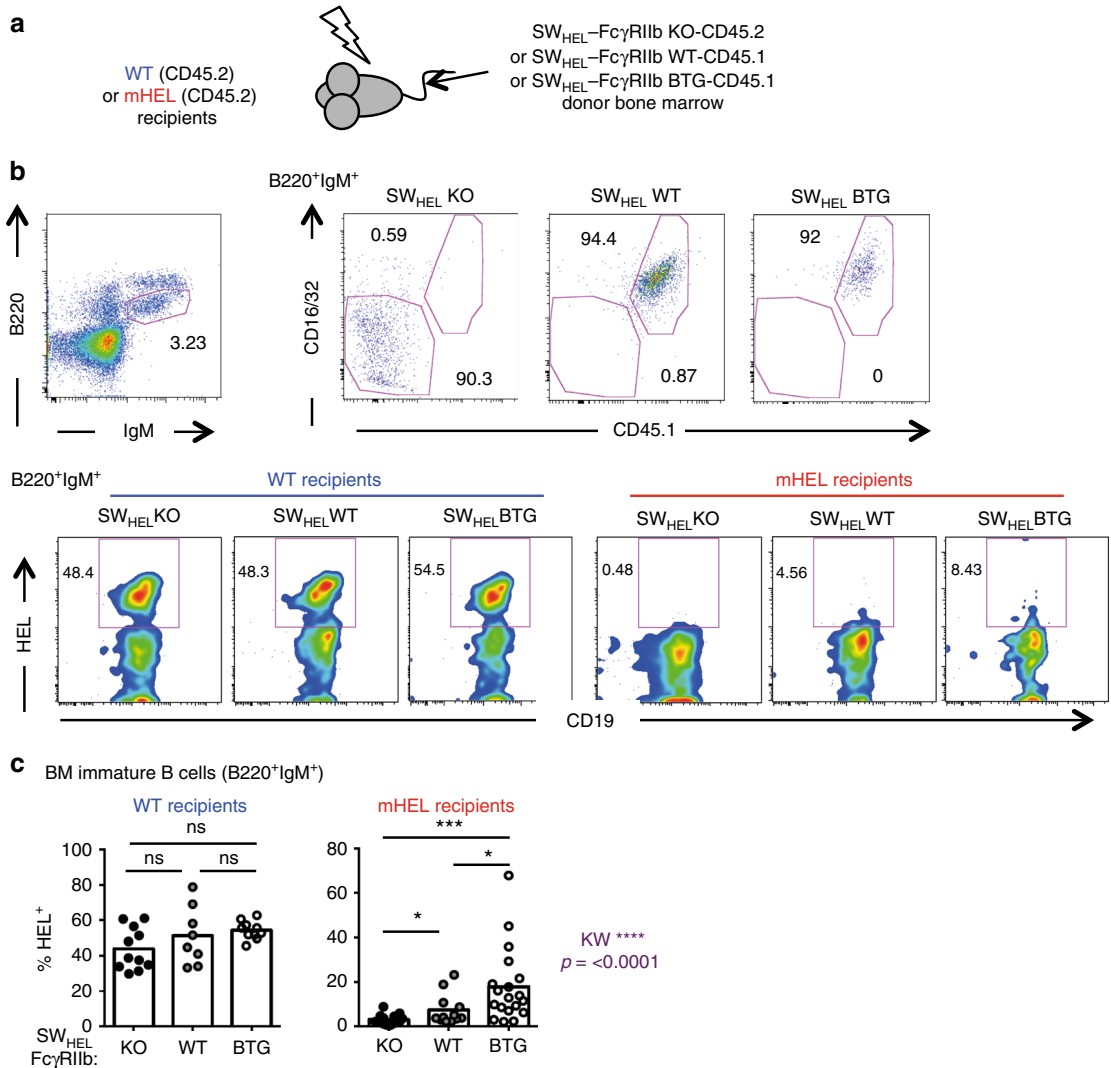

**Fig. 1** Reduced FcγRIIb expression increases central tolerance. **a** Experimental procedure: wild-type and mHEL recipients (both CD45.2) were lethally irradiated and reconstituted with bone marrow from $SW_{HEL}$-FcγRIIb KO-CD45.2, $SW_{HEL}$-FcγRIIb WT-CD45.1 or $SW_{HEL}$-FcγRIIb BTG-CD45.1 donor mice. **b** Representative flow plots of immature B cells in the bone marrow (B220$^+$IgM$^+$, left top panel). B cells originating from the donor bone marrow were identified based on FcγRIIb expression (detected with an anti-CD16/32 Ab) and expression of the congenic marker CD45.1 (right top panels). The frequency of HEL-specific cells for each condition is showed using the smoothing function to help the visualisation of rare events (bottom panels). **c** Quantification of the frequency of HEL-specific immature B cells in WT (left panel) and mHEL (right panel) recipient mice. mHEL recipients: $n = 12$–19 mice per group; wt recipients: $n = 8$–11 mice per group. The mean is represented and each dot corresponds to an individual mouse. Four pooled experiments are shown. The p-values were determined with the Kruskal–Wallis (KW) non-parametric test when comparing the three groups (KW: the p-value is indicated in purple for the mHEL recipient. All KW tests were not significant for WT recipients) and with the Mann–Whitney non-parametric test when two conditions were compared (indicated by black stars). *$p < 0.05$; **$p < 0.01$; ***$p < 0.001$. Comparisons generating non-significant p-values are indicated with "ns". Source data are provided as a Source Data file

expression was associated with more stringent counter-selection of HEL-specific B cells at all stages from transitional T1 to mature follicular and MZ B cells when compared with WT controls (Fig. 2b–g). Supporting this observation, B cell-specific over-expression of FcγRIIb was associated with increased frequency of HEL-specific B cells in mHEL mice reconstituted with $SW_{HEL}$-FcγRIIb BTG BM cells compared with KO, and in some populations WT, groups, consistent with reduced tolerance at the transitional T2/T3 (Fig. 2d) and mature follicular (Fig. 2e, f) B-cell stages. Low but detectable levels of anti-HEL IgG1 were seen in sera of unimmunised mHEL mice reconstituted with BM from all $SW_{HEL}$ mice (Supplementary Fig. 1d). This observation makes it possible that the effect of FcγRIIb on selection at this stage might be driven by co-cross-linking of the BCR and FcγRIIb with cognate antibody, although we cannot exclude an effect of

FcγRIIb on tonic BCR signalling independently of antigen recognition and of FcγRIIb co-crosslinking[46,47].

To confirm the role of FcγRIIb in clonal deletion at these central and peripheral tolerance checkpoints, we crossed the $SW_{HEL}$-FcγRIIb KO-CD45.2 and the $SW_{HEL}$-FcγRIIb BTG-CD45.1 mouse strains with mHEL mice (Supplementary Fig. 2). In this experimental model, in contrast to BM chimeras where HEL is only expressed by radiation-resistant recipient cells, all cells including hematopoietic cells express the membrane-bound HEL autoantigen. We observed a reduced frequency of HEL-specific B cells in $SW_{HEL}$-FcγRIIb-KO × mHEL compared with $SW_{HEL}$-FcγRIIb-BTG × mHEL mice (Supplementary Fig. 2b–e, right panels), suggesting that reduced expression of FcγRIIb leads to more potent clonal deletion regardless of which cells express the autoantigen. Taken together, these results show that absence

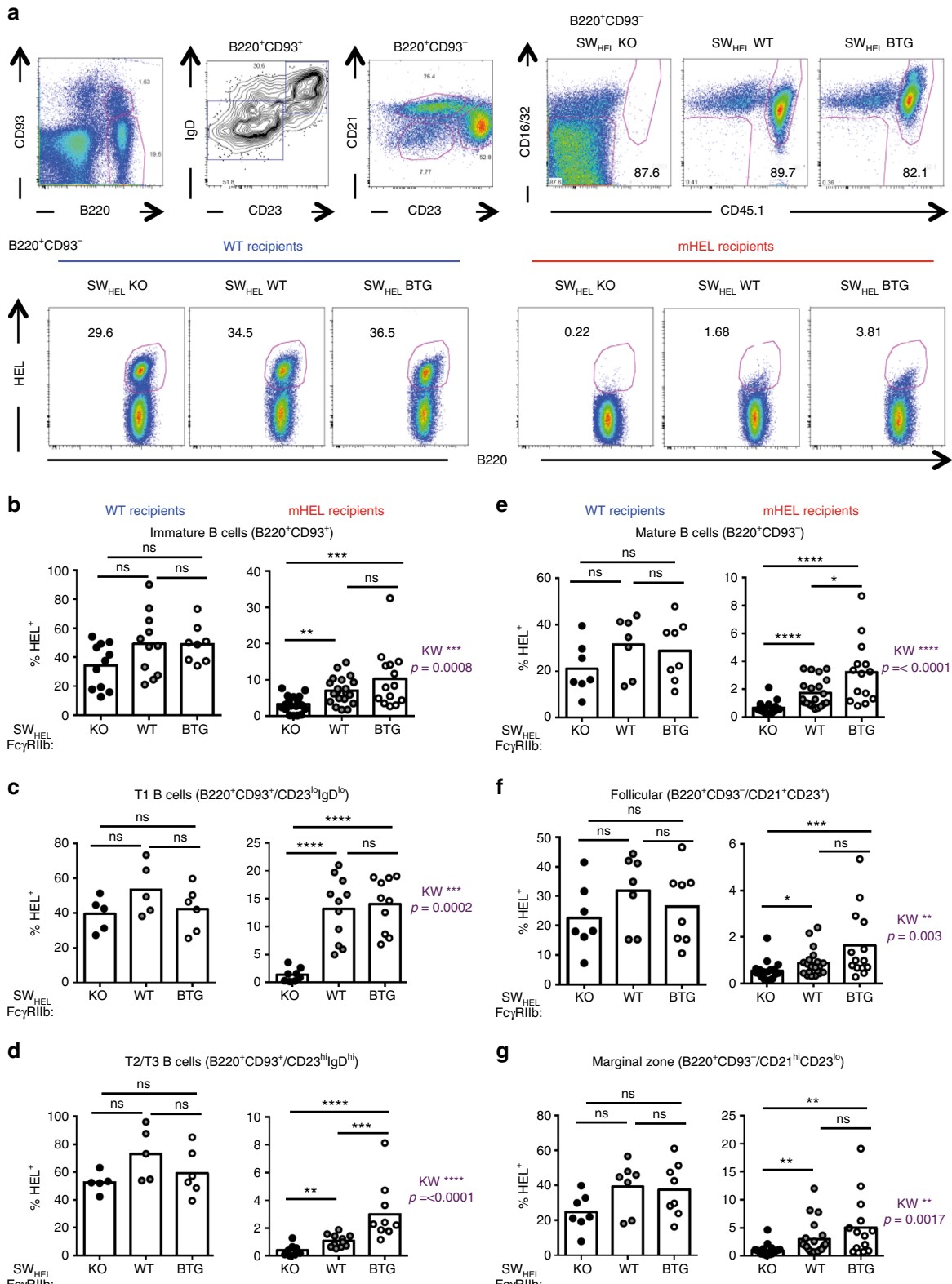

of or reduced FcγRIIb expression enhances clonal deletion at the central and transitional B-cell peripheral tolerance checkpoints, raising the question of how FcγRIIb deficiency leads to autoimmunity.

**FcγRIIb limits autoreactive B-cell anergy**. We therefore tested whether FcγRIIb expression controls B-cell anergy[10,13,45] in

addition to clonal deletion. We measured anergy by *ex vivo* cross-linking of the BCR and analysis of the phosphorylation of a downstream kinase, normally reduced in anergic B cells. The intensity of the phospho-Syk staining was equivalent between HEL-specific (HEL+) and non-HEL-specific B cells (HEL−) in WT recipients independent of FcγRIIb expression, that is, the ratio of the geometric mean expression of phospho-Syk in HEL+ to HEL− B cells was 1 (Fig. 3a, b). This ratio was less than 1 in

**Fig. 2** Reduced FcγRIIb expression increases peripheral tolerance at the transitional B-cell stage. **a** Representative flow plots of immature and mature B cells in the spleen: immature and mature B cells were identified as $B220^+CD93^+$ and $B220^+CD93^-$, respectively. Amongst the immature B cells, transitional T1 and T2/T3 were identified based on the expression of IgD and CD23 (T1 = $IgD^{lo}CD23^{lo}$; T2/T3: $IgD^{hi}CD23^{hi}$). Mature B cells were separated as follicular ($CD23^+CD21^+$) and marginal zone ($CD23^{lo}CD21^{hi}$) (left top three panels). On each subset, B cells originating from the donor bone marrow were identified based on the expression of FcγRIIb (CD16/32) and of the congenic marker CD45.1 (right top three panels). The frequency of HEL-specific cells was determined (bottom panels). **b–g** Quantification of the frequency of HEL-specific B cells in WT (left panel) and mHEL (right panel) recipient mice: total immature B cells (**b**), T1 transitional B cells (**c**), T2/T3 transitional B cells (**d**), total mature B cells (**e**), follicular B cells (**f**) and marginal zone B cells (**g**). mHEL recipients: $n = 14$–20 mice per group; wt recipients: $n = 7$–8 mice per group. The mean is represented and each dot corresponds to an individual mouse. Four pooled experiments are shown. The p-values were determined with the Kruskal–Wallis (KW) non-parametric test when comparing the three groups (KW: the p-value is indicated in purple for the mHEL recipient. All KW tests were not significant for the WT recipients) and with the Mann–Whitney non-parametric test when two conditions were compared (indicated by black stars). *$p < 0.05$; **$p < 0.01$; ***$p < 0.001$; ****$p < 0.0001$. Comparisons generating non-significant p-values were indicated with "ns". Source data are provided as a Source Data file

mHEL mice reconstituted with $SW_{HEL}$-FcγRIIb-WT BM, consistent with increased anergy among the remaining HEL+ B cells that have avoided deletion (Fig. 3a, b). This ratio was further reduced to 0.5 for mice reconstituted with $SW_{HEL}$-FcγRIIb-KO BM, demonstrating reduced phosphorylation of Syk upon BCR engagement on HEL-autoreactive B cells in absence of FcγRIIb, and conversely was around 1.5 for mHEL mice reconstituted with $SW_{HEL}$- FcγRIIb-BTG BM (Fig. 3a, b). Moreover, autoreactive HEL+ B cells proliferated less than HEL− B cells in response to LPS as shown by reduced CFSE dilution (Fig. 3c, d), and this reduced proliferation was more marked in the absence of FcγRIIb, suggesting that, indeed, FcγRIIb expression controls autoreactive B-cell anergy (Fig. 3c, d). Consistent with this, the frequency of HEL+ plasmablasts ($CD138^+B220^{lo}$) was higher in mHEL recipients reconstituted with $SW_{HEL}$-FcγRIIb-BTG and $SW_{HEL}$-FcγRIIb-WT than with $SW_{HEL}$-FcγRIIb-KO BM (Fig. 3e, f). Hence, our results show that absence of FcγRIIb expression promotes increased anergy, with reduced signalling, proliferation and differentiation of HEL-specific autoreactive B cells.

**FcγRIIb limits autoreactive GC response against mHEL.** Thus FcγRIIb expression regulates pre-immune tolerance, with FcγRIIb-deficient mice showing enhanced deletion and anergy compared with their control counterparts. This is at odds with the observation that FcγRIIB defects are associated with autoimmunity in both mice and humans, raising the question of which tolerance checkpoint is defective in absence of FcγRIIB.

We therefore assessed GC tolerance, generating chimeras as above and inducing GCs by immunising with HEL conjugated to sheep red blood cells (SRBC-HEL) (Fig. 4a). Following SRBC-HEL immunisation of mHEL recipients reconstituted with $SW_{HEL}$-FcγRIIb-KO BM cells, the frequency and number of HEL-specific follicular B cells were reduced (Fig. 2, Supplementary Figs. 1 and 3, as expected, due to enhanced pre-immune tolerance) but despite this, HEL-specific GC B-cell number was increased compared with $SW_{HEL}$-FcγRIIb-WT and $SW_{HEL}$-FcγRIIb-BTG controls (Fig. 4b–d). The ratio of the frequencies of GC versus follicular HEL+ B cells was significantly increased in mHEL mice reconstituted with $SW_{HEL}$-FcγRIIb-KO BM cells compared with the two other experimental groups, emphasising the large expansion of HEL-autoreactive GC B cells in the absence of FcγRIIb (Fig. 4e). Despite this, the frequency and number of $T_{FH}$ and $T_{FR}$ cells were not significantly different (Supplementary Fig. 4). To determine if this increase in HEL-specific autoreactive GC B cells led to the development of HEL-specific plasma cells, we performed ELIspot, and observed more HEL-specific antibody-forming cells (AFCs) in the spleen of mHEL recipients reconstituted with $SW_{HEL}$-FcγRIIb-KO BM compared with the the $SW_{HEL}$-FcγRIIb-WT and -BTG groups (Fig. 4f). The size and intensity of the spots were similar between the different experimental conditions (Supplementary Fig. 5), consistent with

equivalent antibody output and in vitro survival. Thus, despite a mature B-cell repertoire shaped by enhanced clonal deletion and anergy at pre-immune tolerance checkpoints, mice deficient in FcγRIIb have a marked defect in the negative-selection of autoreactive GC B cells.

**FcγRIIb limits autoreactive GC response against sHEL.** To assess if the influence of FcγRIIb on B-cell tolerance was dependent on the strength of BCR signalling, we next used mL5 transgenic mice that express soluble HEL, which binds the BCR with less avidity than mHEL[2], as an autoantigen (Fig. 5a). As expected from the previous studies[2], the deletion of HEL-specific B cells was less pronounced in mL5 recipients compared with mHEL recipients (Figs. 1, 2 and 5). Nevertheless, deficient FcγRIIb expression still reduced the frequency of HEL-specific BM immature B cells in mice reconstituted with $SW_{HEL}$-FcγRIIb-KO compared with $SW_{HEL}$-FcγRIIb-WT or $SW_{HEL}$-FcγRIIb-BTG BM (Fig. 5b). The frequency and number of HEL-specific autoreactive B cells in the marginal zone were also decreased in absence of FcγRIIb (Fig. 5d) compared with $SW_{HEL}$-FcγRIIb-BTG BM, whereas the frequency and number of autoreactive non-GC follicular B cells was not significantly affected by FcγRIIb expression (Fig. 5c).

Following immunisation, we observed an increase in the number of HEL-specific GC B cells (Fig. 5e) and in the ratio of HEL-specific GC to follicular B cells (Fig. 5f) in sHEL recipient mice reconstituted with $SW_{HEL}$-FcγRIIb-KO BM compared with mice reconstituted with $SW_{HEL}$-FcγRIIb-BTG BM. Finally, sHEL recipients reconstituted with $SW_{HEL}$-FcγRIIb-KO BM had an increased number of HEL-specific AFCs compared with the two other groups (Fig. 5g). Thus, whether autoantigen is soluble or membrane-bound, FcγRIIb deficiency enhances pre-immune and reduces GC tolerance.

**FcγRIIb controls bystander autoreactive GC B cells.** We previously generated a knock-in mouse model bearing naturally occurring polymorphisms of the *Fcgr2b* promoter common in wild mice and associated with spontaneous autoimmunity in inbred strains. These $FcγRIIb^{wild/H1}$ KI mice have defective upregulation of FcγRIIb upon activation, leading to an increase in total GC B cell number, but no change in the number of antigen-specific GC B cells after a T-dependent immunisation[44]. This suggested that FcγRIIb expression might regulate the number of bystander GC B cells, that is, those displaying no or very low affinity for the immunising antigen. Moreover, the $FcγRIIb^{wild/H1}$ KI mice also produced anti-dsDNA and anti-chromatin autoantibodies transiently following exogenous antigenic challenge[44]. Consistent with this, SRBC immunised FcγRIIb-deficient mice produced significantly more anti-dsDNA autoantibodies than BTG mice (Fig. 6a). These findings raised the possibility that FcγRIIb might contribute to GC tolerance by restricting the

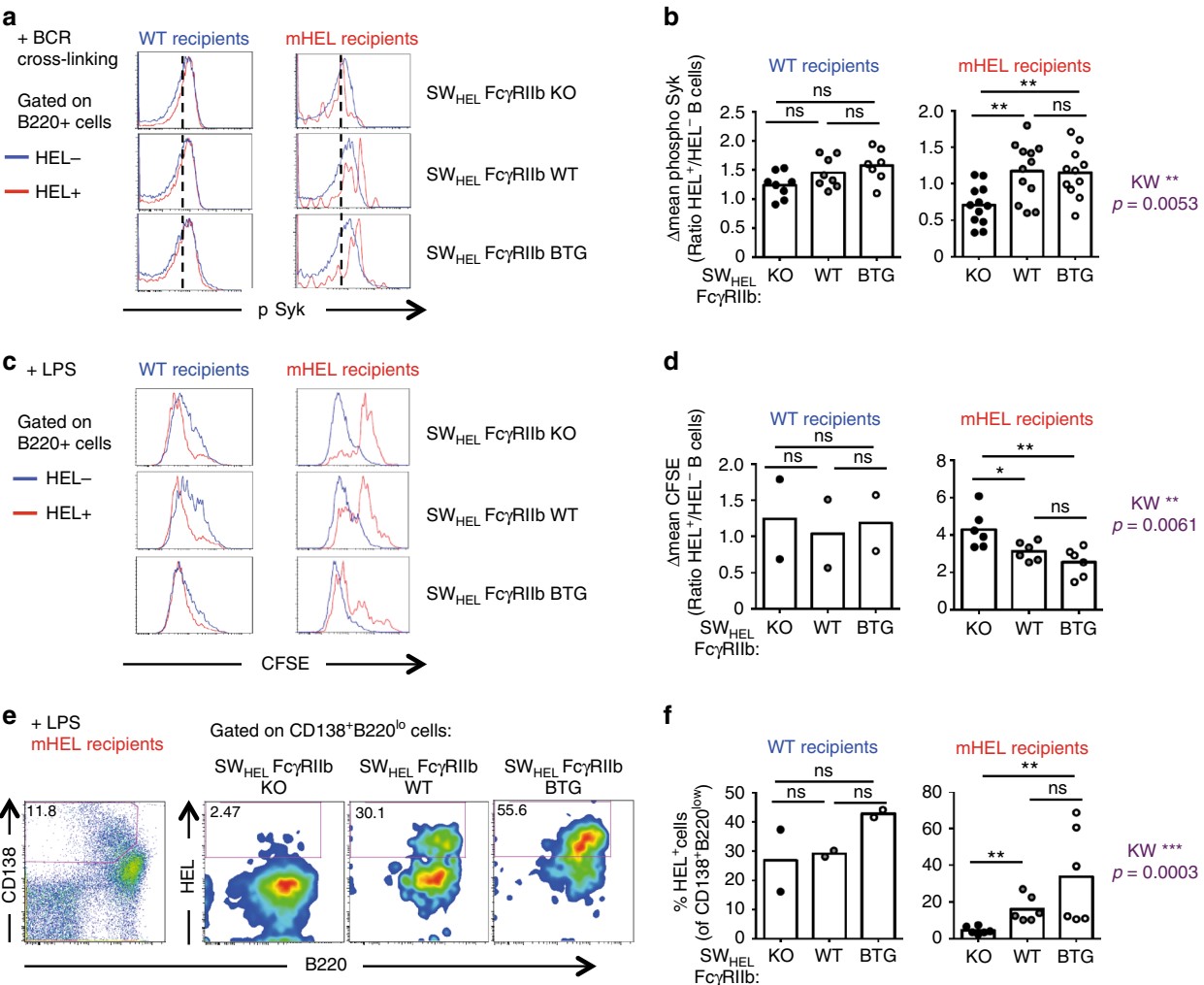

**Fig. 3** Absence of FcγRIIb expression enhances autoreactive B-cell anergy. **a** Representative histograms of phospho-Syk on HEL-specific (HEL+, red histograms) and HEL-non specific (HEL−, blue histograms) splenic B cells from WT and mHEL recipient chimeras reconstituted with SW$_{HEL}$-FcγRIIb KO, SW$_{HEL}$-FcγRIIb WT and SW$_{HEL}$-FcγRIIb BTG BM and measured by flow cytometry after BCR cross-linking. A dashed line indicates the staining without stimulation. **b** Quantification of phospho-Syk after cross-linking of the BCR. For each sample, the geometric mean of the phospho-Syk staining without stimulation was subtracted from the one after stimulation. We used this corrected mean to calculate for each mouse the ratio of phospho-Syk staining in HEL+ by HEL− B cells (Δmean phospho-Syk). **c** Representative histograms of CFSE dilution on HEL-specific (HEL+, red) and HEL-non specific (HEL−, blue) splenic B cells from WT and mHEL recipient chimeras reconstituted with SW$_{HEL}$-FcγRIIb KO, SW$_{HEL}$-FcγRIIb WT and SW$_{HEL}$-FcγRIIb BTG BM 4 days after LPS stimulation. **d** CFSE dilution was quantified by flow cytometry on HEL+ and HEL− B cells. For each mouse, we calculated the ratio of the geometric mean of the CFSE staining on HEL+ by HEL− B cells (Δmean CFSE). **e** Representative gating of plasmablasts (left panel) and HEL+ plasmablasts (right panels) from mHEL recipient chimeras reconstituted with SW$_{HEL}$-FcγRIIb KO, SW$_{HEL}$-FcγRIIb WT and SW$_{HEL}$-FcγRIIb BTG BM 4 days after LPS stimulation. **f** B-cell differentiation after LPS stimulation was measured by quantifying the frequency of HEL+ plasmablasts. For **a**, **b**, three experiments were pooled. Wild-type recipients: $n = 7$–8 mice/group; mHEL recipients: $n = 11$–12 mice/group. For **c**–**e**, two experiments were pooled. Wild-type recipients: $n = 2$ mouse/group; mHEL recipients: $n = 6$ mice/group. The mean is represented and each dot corresponds to an individual mouse. The p-values were determined with the Kruskal–Wallis (KW) non-parametric test when comparing the three groups (KW: the p-value is indicated in purple for the mHEL recipient. All KW tests were not significant for WT recipients) and with the Mann–Whitney non-parametric test when two conditions were compared (indicated by black stars). *$p < 0.05$; **$p < 0.01$; ***$p < 0.001$. Comparisons generating non-significant p-values were indicated with "ns". Source data are provided as a Source Data file

generation of autoreactive cells arising as bystanders in a GC reaction to non-self antigens.

We then asked whether, in the absence of FcγRIIb expression, autoreactive bystander B cells could expand in the GC in response to exogenous antigen and generate autoantibodies. We crossed the mHEL strain to the $Rag2^{-/-}$ background (ensuring all B cells were of BM donor origin, allowing precise quantification of rare HEL-specific GC B cells and serum HEL-specific antibodies), irradiated both $Rag2^{-/-}$ and $Rag2^{-/-}$ mHEL mice sub-lethally, and reconstituted them with SW$_{HEL}$-FcγRIIb-KO, -WT or -BTG BM

cells (Fig. 6b). As expected, enhanced pre-immune tolerance resulted in fewer HEL-specific B cells in mHEL recipient mice reconstituted with SW$_{HEL}$-FcγRIIb-KO compared with SW$_{HEL}$-FcγRIIb-WT and SW$_{HEL}$-FcγRIIb-BTG BM cells (Fig. 6c). However, SRBC immunisation drove an increase in HEL-specific GC B cells (Fig. 6d) and in serum HEL-specific IgG1 (Fig. 6e) in $Rag2^{-/-}$ mHEL mice reconstituted with SW$_{HEL}$-FcγRIIb-KO BM compared with SW$_{HEL}$-FcγRIIb-WT and -BTG BM cells.

We next wondered if this bystander autoimmunity could be observed in a more physiological context, and thus performed

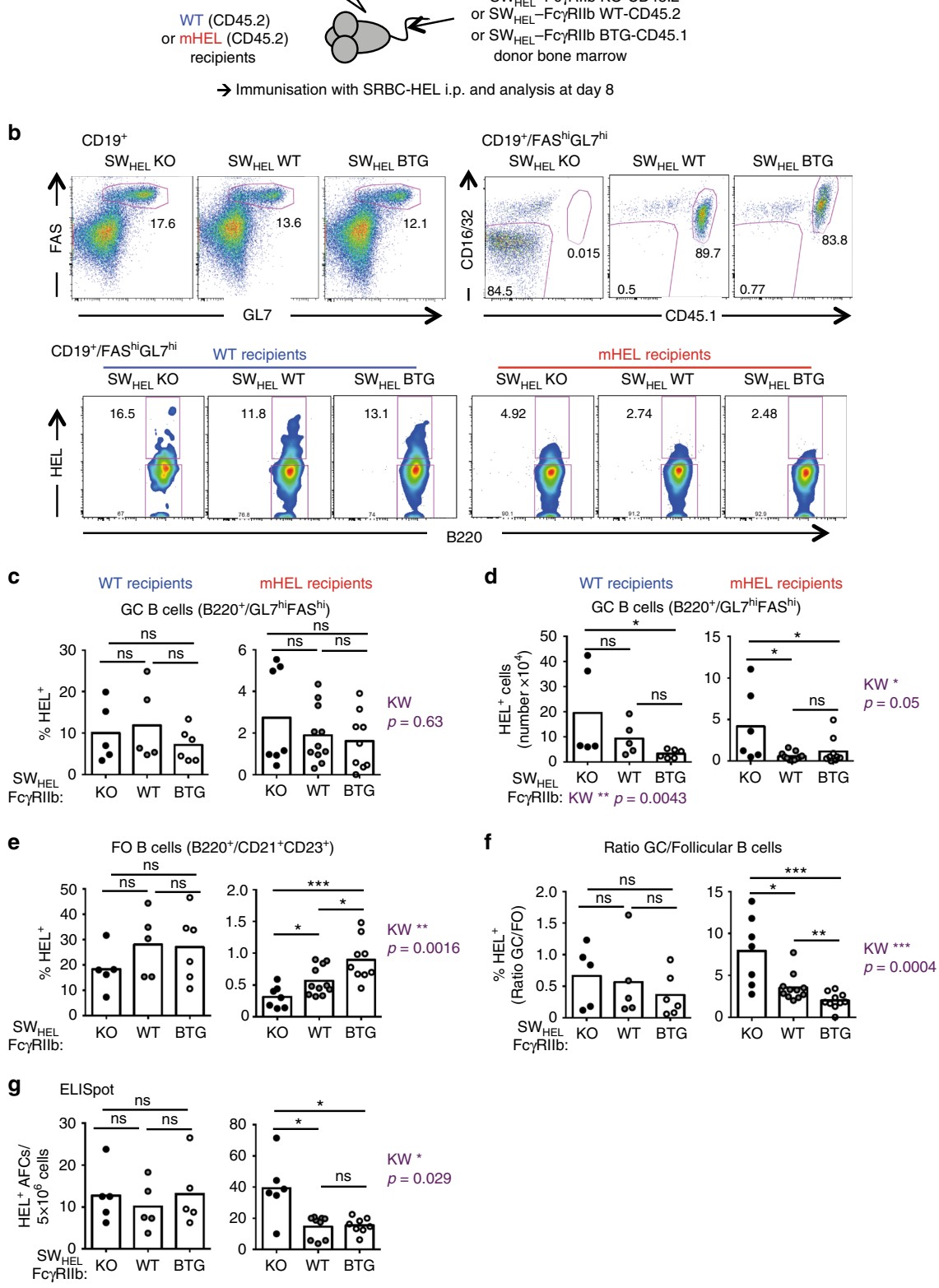

similar experiments with the SW_HEL-FcγRIIb^wild/H1 KI mice containing naturally occurring promoter *Fcgr2b* variations (Supplementary Fig. 6a). We had shown previously that these mice displayed subtly reduced FcγRIIb expression on pre-B, immature and GC B cells, suggesting these natural *Fcgr2b* polymorphisms may impact upon both pre-immune and GC tolerance[44]. The results followed the same trend as those

generated using SW_HEL-FcγRIIb KO chimeras. There was enhanced tolerance at the pre-immune checkpoints (reduced frequency of HEL-specific autoreactive B cells in the BM and spleen of SW_HEL-FcγRIIb^wild/H1 KI/mHEL mice compared with control (Supplementary Fig. 6b, c). Despite this, in the SW_HEL-FcγRIIb^wild/H1 KI chimeric mice immunised with SRBC, the total number of HEL-specific GC B cells as well as the ratio of HEL-

**Fig. 4** FcγRIIb limits autoreactive GC response despite more stringent pre-immune tolerance. **a** Wild-type and mHEL recipients (both CD45.2) were irradiated and reconstituted with the bone marrow from SW$_{HEL}$-FcγRIIb KO-CD45.2, SW$_{HEL}$-FcγRIIb WT-CD45.1 or SW$_{HEL}$-FcγRIIb BTG-CD45.1 donor mice. Post reconstitution, chimeras were immunised with SRBC-HEL intraperitoneally and analysed 8 days later. **b** Representative flow plots of GC B cells (top panels) in the spleen of immunised chimeric mice. GC B cells were gated as CD19$^+$/FAS$^{hi}$GL7$^{hi}$. B cells originating from the donor bone marrow were detected by the level of FcγRIIb expression (CD16/32) and by the congenic marker CD45.1 (top right three panels) and the frequency of HEL-specific cells was measured (bottom panels). **c, d** Quantification of the frequency and number of HEL-specific GC B cells in WT (left panels) and mHEL (right panels) recipient mice. **e** Quantification of the frequency of HEL-specific follicular B cells in WT (left panels) and mHEL (right panels) recipient mice. **f** Ratio of the frequency of HEL-specific GC B cells and total follicular B cells in WT (left panel) and mHEL recipient mice (right panel). **g** Quantification by ELISpot of HEL-specific antibody-forming cell (AFC) numbers in the spleen of immunised WT (left panels) and mHEL (right panels) recipient mice. For all panels two experiments were pooled. For panels (**c–f**), wild-type recipients: $n = 5$–6 mice per group; mHEL recipients: $n = 8$–11 mice per group. For panel (**g**), wild-type recipients: $n = 5$ mice per group; mHEL recipients: $n = 6$–9 mice per group. The mean is represented and each dot corresponds to an individual mouse. In panel **b**, the FlowJo smoothing option was used to improve visibility due to low event rates. The p-values were determined with the Kruskal–Wallis (KW) non-parametric test when comparing the three groups (KW: the p-value is indicated in purple for the mHEL recipient. All KW tests were not significant for WT recipients with the exception of panel d, for which the KW p-value is indicated for the WT group as well) and with the Mann–Whitney non-parametric test when two conditions were compared (indicated by black stars). *$p < 0.05$; **$p < 0.01$; ***$p < 0.001$. Comparisons generating non-significant p-values were indicated with "ns". Source data are provided as a Source Data file

specific GC B cells to total splenic B cells were increased (Supplementary Fig. 6d, e). A non-significant trend towards increased serum HEL-specific autoantibodies in mice reconstituted with SW$_{HEL}$-FcγRIIb$^{wild/H1}$ KI BM was also observed (Supplementary Fig. 6f). In summary, a naturally occurring genetic variant reducing the FcγRIIb increase that follows B-cell activation can drive the expansion of autoreactive B cells in the GC, despite enhanced pre-immune tolerance and even in response to an irrelevant exogenous antigen.

**FcγRIIB limits VH4-34 gene usage in humans.** We then asked whether FcγRIIB had a similar impact on tolerance checkpoints in humans. We used the National Institute of Health Research (NIHR) Cambridge BioResource to recruit 29 healthy volunteers: 19 volunteers homozygous for isoleucine at position 232 in *FCGR2B* (hereafter referred to as I232 individuals), the common variant associated with normal FcγRIIB function, and 10 volunteers homozygous for the SLE-associated single-nucleotide polymorphism in *FCGR2B* (hereafter referred to T232 individuals) encoding a receptor that has markedly reduced inhibitory function[21,22]. We flow-sorted peripheral blood CD19$^+$IgD$^+$CD27$^-$ naive/transitional B cells; CD19$^+$IgD$^-$CD27$^-$CD38$^{mid/hi}$ activated B cells; CD19$^+$IgD$^+$CD27$^+$ B cells that include marginal zone B cells, and perhaps B1-like and unswitched memory cells[48–51]; CD19$^+$IgD$^-$CD27$^+$CD38$^{low/mid}$ memory B cells; and CD19$^+$IgD$^-$CD27$^+$CD38$^{hi}$ plasmablasts (Supplementary Fig. 7). The BCR repertoire for each of these subsets was analysed by high-throughput sequencing using a method that can distinguish between isotype classes (Supplementary Fig. 8 and [52]), generating an average of 20,573 BCR sequences per B-cell subset (Supplementary Table 1).

No significant differences in the proportion of B cells within each B-cell subset, nor of isotype usage, was seen between genotypes (Supplementary Figs. 9 and 10). The effect of FcγRIIB on tolerance was evaluated by examining the frequency of cells expressing the *IgHV4-34* gene. This specific heavy chain has been shown to bind red blood cell antigens[53,54] as well as commensal bacteria[55], and was significantly enriched in SLE patients (Fig. 7a) in agreement with previous studies[56,57]. Moreover, *IgHV4-34* was enriched among BCRs without sequence evidence of switching or SHM (comprising predominantly naive B cells), as well as antigen-experienced BCRs as evidenced by isotype switching or SHM, suggesting a defect in both pre- and post-immune tolerance (Fig. 7a).

In I232 healthy volunteers, the frequency of *IGHV4-34* gene usage was higher in CD19$^+$IgD$^+$CD27$^-$ than in CD19$^+$IgD$^-$CD27$^-$CD38$^{mid/hi}$ and CD19$^+$CD27$^+$IgD$^-$ populations (Fig. 7b), consistent with the impact of peripheral tolerance on

this autoreactive population. Healthy T232 and I232 individuals had similar *IGHV4-34* proportions in CD19$^+$IgD$^+$CD27$^-$ cells, suggesting the enhanced central tolerance seen in FcγRIIB-deficient mice may not occur in humans. *IGHV4-34* was, however, enriched in CD19$^+$CD27$^+$IgD$^+$ and CD19$^+$IgD$^-$CD27$^-$CD38$^{mid/hi}$ populations to a level similar to that seen in SLE patients (Fig. 7b), and consistent with impaired post-immune tolerance associated with FcγRIIB dysfunction, though this difference was not seen in the CD19$^+$IgD$^-$CD27$^+$CD38$^{low/mid}$ population. These human data suggest that, even in a healthy heterogeneous population, reduced FcγRIIB function impacts upon post-immune, rather than central/pre-immune tolerance, consistent with the differential impact of FcγRIIB on pre- and post-immune tolerance seen in the mouse, but with notable species-specific differences.

## Discussion

Immune tolerance in B cells is governed by a number of checkpoints that share the common goal of preventing the emergence of autoreactive lymphocytes. Given this shared goal, it might seem likely that molecules that regulate tolerance would impact upon different tolerance checkpoints in a similar direction. One such molecule is FcγRIIb, an inhibitory receptor that antagonises BCR signalling. Its reduced expression or function enhances BCR signalling and also predisposes to autoimmunity in mouse and human. To investigate the role of FcγRIIb in the development of autoimmunity, we used mice either under- or overexpressing FcγRIIb on their B cells, crossed to the SW$_{HEL}$ mouse model to allow interrogation of specific tolerance checkpoints.

Three distinct mouse models were used in this study, as each alone had inherent limitations. The FcγRIIb-deficient mouse has a profound defect affecting not only B cells, but also other cells implicated in the immune response[35,37]. We therefore also used two additional models. In one FcγRIIb was over-expressed on all B cells by between two- and tenfold (but was expressed at normal levels on other cell types[40]). The other is a more physiological model of FcγRIIb under-expression, in which a knock-in of a naturally occurring promoter variant results in a failure of activation-induced upregulation of FcγRIIb specifically on B cells[44]. This genetic model would be expected to give a less extreme phenotype than the deficient mice, as its defect is subtle, but would confirm the B-cell specificity of any effect. Taken together comparisons between these three models increase confidence in our findings, as they provide confirmation in two under-expression models—one an extreme knockout, and one a subtle B-cell specific physiological genetically modified mouse—and a cell-specific overexpression model.

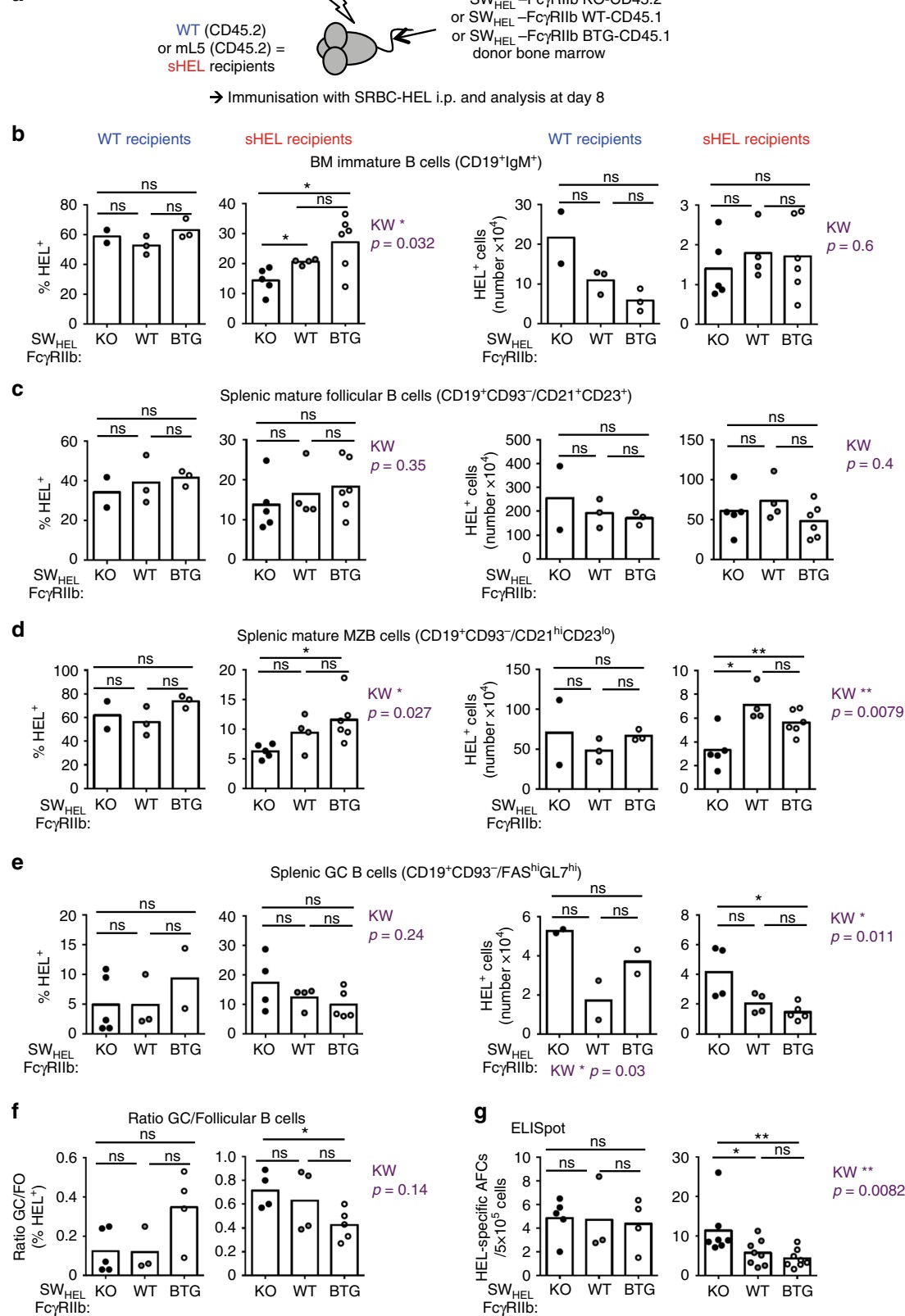

In both major pre-immune tolerance checkpoints, at the BM immature B-cell stage and at the transitional B-cell stage in the spleen, no or low-FcγRIIb expression was associated with more stringent tolerance, with increased deletion and anergy of HEL-specific autoreactive B cells. In contrast, such reduced FcγRIIb expression was associated with the increased production of

autoreactive B cells in the GC, and in serum autoantibody, consistent with a failure of tolerance at the GC checkpoint.

FcγRIIB is expressed not only on B cells, but on other hemato-poietic cells, such as macrophages and dendritic cells. Chimeras reconstituted with FcγRIIb KO BM will therefore contain non-B cells that are FcγRIIb deficient. The impact of FcγRIIb on tolerance,

**Fig. 5** The impact of FcγRIIb on GC tolerance is also observed with a soluble autoantigen. **a** Wild type and sHEL recipients (both CD45.2) were irradiated and reconstituted with bone marrow from SW_HEL-FcγRIIb KO-CD45.2, SW_HEL-FcγRIIb WT-CD45.1 or SW_HEL-FcγRIIb BTG-CD45.1 donor mice. Post reconstitution, chimera were immunised with SRBC-HEL intraperitoneally and analysed 8 days later. All populations were gated as in Figs. 1, 2 and 4. **b-e** Quantification of the frequency of HEL-specific B cells in WT (left panels) and sHEL (right panels) recipient mice: BM immature B cells (**b**), splenic mature MZ B cells (**c**), splenic follicular B cells (**d**) and splenic GC B cells (**e**). **f** Ratio of HEL-specific GC B cells and total follicular B cells in WT (left panel) and sHEL recipient mice (right panel). **g** Quantification by ELISpot of HEL-specific antibody-forming cells (AFCs) in the spleen of immunised WT (left panels) and sHEL (right panels) recipient mice. sHEL recipients: $n = 4-6$ mice per group; wt recipients: $n = 2-3$ mice per group. One experiment representative of at least two is shown. The mean is represented and each dot corresponds to an individual mouse. The p-values were determined with the Kruskal–Wallis (KW) non-parametric test when comparing the three groups (KW: the p-value is indicated in purple for the mHEL recipient. All KW tests were not significant for WT recipients with the exception of panel **e** (absolute number) for which KW p-value is indicated for the WT group as well) and with the Mann–Whitney non-parametric test when two conditions were compared (indicated by black stars). *$p < 0.05$; **$p < 0.01$. Comparisons generating non-significant p-values were indicated with "ns". Source data are provided as a Source Data file

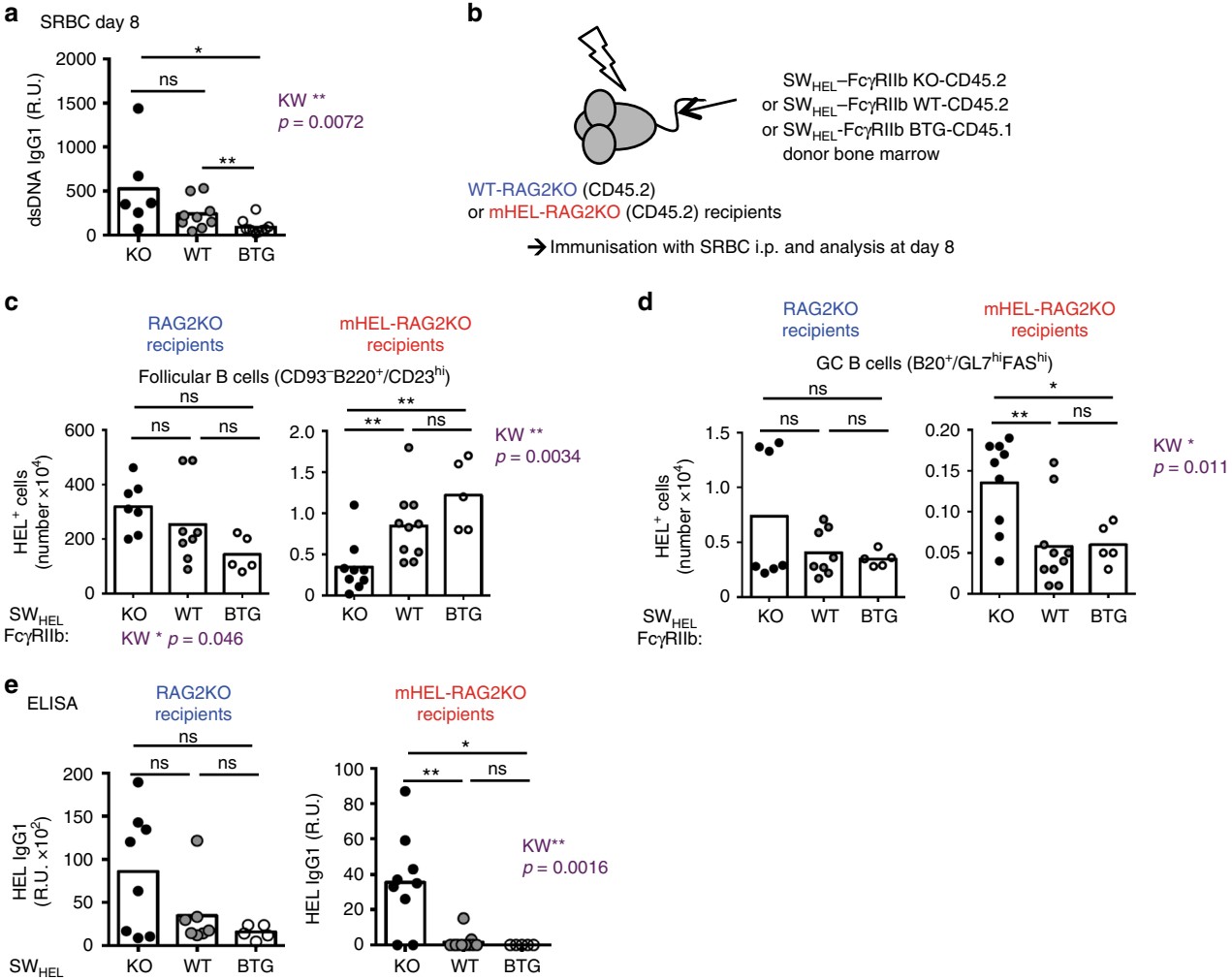

**Fig. 6** FcγRIIb expression controls bystander autoreactive GC B-cell expansion. **a** Serum dsDNA-specific IgG1 titer in KO-CD45.2, WT-CD45.2 or BTG-CD45.1 mice immunised for 8 days with SRBC. (R.U. = relative units) **b** RAG2KO and RAG2KO-mHEL recipients (both CD45.2) were irradiated and reconstituted with bone marrow from SW_HEL-FcγRIIb KO-CD45.2, SW_HEL-FcγRIIb WT-CD45.2 or SW_HEL-FcγRIIb BTG-CD45.1 donor mice. Post reconstitution, chimeras were immunised with SRBC intraperitoneally and analysed 8 days later. **c** Number of HEL-specific follicular B cells in RAG2KO (left panel) and RAG2KO-mHEL (right panel) recipient mice. **d** Number of HEL-specific GC B cells in RAG2KO (left panel) and RAG2KO-mHEL (right panel) recipient mice. **e** Serum HEL-specific IgG1 titer in RAG2KO (left panel) and RAG2KO-mHEL (right panel) recipient mice. Two pooled experiments are shown. RAG2KO recipients: $n = 5-8$ mice per group; RAG2KO-mHEL recipients: $n = 5-10$ mice per group. The mean is represented and each dot corresponds to an individual mouse. The p-values were determined with the Kruskal–Wallis (KW) non-parametric test when comparing the three groups (KW: the p-value is indicated in purple for the RAG2KO-mHEL recipient. All KW tests were not significant for RAG2KO recipients with the exception of panel **c** for which KW p-value is indicated for the RAG2KO group as well) and with the Mann–Whitney non-parametric test when two conditions were compared (indicated by black stars). *$p < 0.05$; **$p < 0.01$. Comparisons generating non-significant p-values were indicated with "ns". Source data are provided as a Source Data file

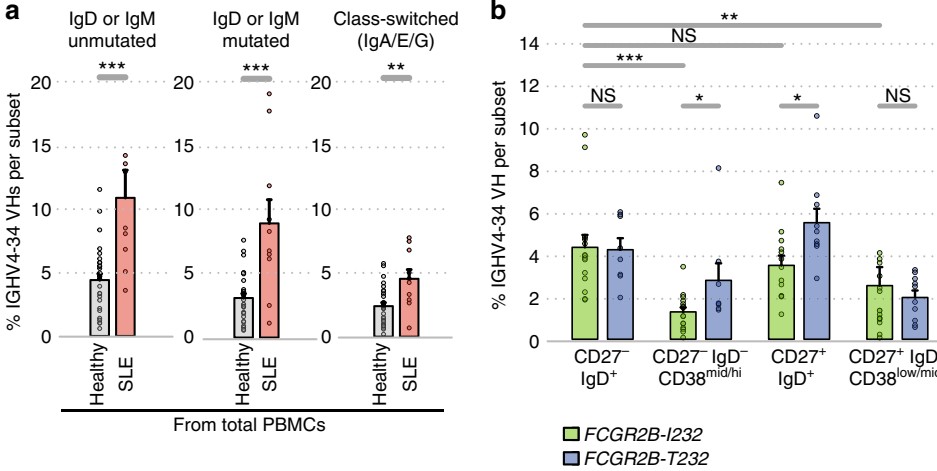

**Fig. 7** Effect of *FCGR2B* genotype on the *IGHV* gene repertoire in humans. **a** Percentage of *IGHV4-34* gene usage from *IgH* sequencing from total PBMCs from healthy individual versus SLE patients during active disease, grouped according to IgH subtype: unmutated IgD or IgM IgH sequences, which are largely derived from naive B cells; mutated IgD or IgM IgH sequences which are derived from B cells having undergone SHM, thus evidence of antigenic stimulation; and class-switched IgH sequences, therefore derived from B cells post-antigenic stimulation. **b** Percentage of *IGHV4-34* gene usage in cell-sorted populations from *FCGR2B* genotyped healthy individuals per population. * denotes differences between genotypes or cell subsets with *p*-values < 0.05, ** denotes *p*-values < 0.005, *** denotes *p*-values < 0.0005 (by ANOVA with age as a covariate), NS: non-significant *p*-values

however, is likely to be B cell intrinsic for several reasons. The first is that we observed a similar effect on tolerance in chimeras reconstituted with BM from SWHEL *Fcgr2b*^*wild/H1*^ KI mice, in which FcγRIIB expression is reduced only on B cells[44]. The second is that transgenic overexpression of FcγRIIb on B cells alone leads to the opposite phenotype, with very strong tolerance at the immature stage despite less stringent GC tolerance. Finally, we also performed our BM reconstitution experiments in *Rag2*^−/−^ recipient mice that were non-lethally irradiated. In consequence, cells other than B and T lymphocytes were mostly coming from the recipients that were WT for FcγRIIb expression.

The two pre-immune checkpoints were similarly influenced by FcγRIIb expression, suggesting a shared mechanism distinct from that in the GC. Co-ligation of FcγRIIb to the BCR by cognate antibody is known to inhibit BCR signalling. It is also clear, however, that in immature B cells FcγRIIb can inhibit the BCR independent of cross-linking by antibody[46]. As we observed detectable levels of anti-HEL IgG1 in relevant mice, it is plausible that they are involved in co-ligation of FcγRIIb to the HEL-specific BCR. It is also possible that FcγRIIb may be having its impact through tonic inhibitory signals. Whether or not the impact of FcγRIIb on selection at the immature/transitional B-cell stage is dependent upon antibody ligation of FcγRIIb, therefore remains an open question.

Our results demonstrate that tolerance in the GC cannot be mediated solely by a deletion of autoreactive clones driven by BCR signal strength as occurs in the BM, as earlier studies might have implied[7,15–17], as the impact of reduced FcγRIIb on BCR signalling should enhance such tolerance. By demonstrating FcγRIIb deficiency increased HEL-specific GC B cells in response to SRBC in mice that had never been exposed to HEL, we confirmed our earlier suspicion[44] that FcγRIIb controlled the expansion of bystander B cells in the GC. Such bystander cells, with no measurable affinity for the immunising antigen, have recently been implicated in contributing to ongoing GCs[58,59], supporting our observation. It should be noted, however, that while low affinity cross-reactivity between the transgenic anti-HEL BCR and SRBC antigens cannot be measured, and would seem unlikely, the presence in the bystander population of cells with very low affinity for a given antigen cannot be excluded, and could contribute to subsequent measurable bystander responses. Nonetheless, no or

low expression of FcγRIIb led to the presence of autoreactive cells into the GC that could differentiate into AFCs that produced isotype-switched autoantibody detectable in the serum in response to a distinct exogenous antigen. This could explain the appearance of autoantibodies sometimes noted after infection or inflammation in humans[60,61], which is presumably impacted by genetic background. Most importantly, it points to a potential novel mechanism-governing tolerance and the development of autoimmunity in the GC—the control of bystander activation and expansion of autoreactive B cells in the GC could be a central mechanism for avoiding autoimmunity, and one in which inhibitory receptors such as FcγRIIb play an important role. A recent paper by Silver et al. has formally demonstrated, using a mouse transgenic system, that non-cognate B cells can enter the GC and diversify via somatic hypermutation even in a competitive setting, and can then generate new antigenic specificities and ongoing immune responses[58]. This mechanism was proposed to have perhaps arisen to fill holes in the B-cell repertoire, and thus improve defence against infection. Our data suggest it could also be responsible for generating autoreactive responses, and perhaps also for the GC-driven autoantigen spreading recently described by Carroll and colleagues[59]. Thus, the generation of autoimmunity by bystander activation of B cells is consistent with previous observations in the setting of both non-autoreactive and autoreactive immune responses. Further work will be required to determine the precise molecular mechanisms at play.

The results presented here might make the concepts of pre-immune and post-immune tolerance a more logical way of subdividing tolerance than the more traditional central versus peripheral tolerance nomenclature. This constitutes an example of a regulatory receptor having opposing effects on pre- and post-immune B-cell tolerance. It could be envisaged that other molecules that regulate BCR signal strength may also have different impacts upon tolerance checkpoints, as, for example, was reported for a *Ptpn22* risk mutant that differentially regulates autoreactive follicular and marginal zone B cells[62]. The net outcome of FcγRIIb deficiency in the B-cell compartment is autoimmunity, demonstrating that in this particular context post-immune tolerance is dominant over pre-immune tolerance.

This seems also to be the case in humans, as complementary findings were made in healthy volunteers bearing a loss-of-

function variant in *FCGR2B*. The proportion of *IGHV4-34* clones, known to be autoreactive, was significantly higher in the activated CD19$^+$IgD$^-$CD27$^-$CD38$^{mid/hi}$ and CD19$^+$IgD$^+$CD27$^+$ B-cell subsets in individuals homozygous for the *FCGR2B* T232 loss-of-function SLE-associated polymorphism. This is consistent with a relative impairment of post-immune tolerance in people with reduced FcγRIIB function, as was seen in mice. Mouse and human results were not fully concordant (there was, for example, no evidence in humans of increased pre-immune tolerance associated with reduced FcγRIIb function, at least as this is reflected in IGHV4-34 clone frequency). Nonetheless, human data support a predominant role for FcγRIIB in post-immune tolerance, as was seen in the various mouse models.

This unexpected complexity of checkpoint control in B-cell tolerance points to a nuanced regulation of the balance between the risk of autoreactivity and the maintenance of repertoire breadth to optimise defence against infection. It may also have implications for the increasingly sophisticated attempts to manipulate the B-cell immune response following the success of B-cell depletion by rituximab in autoimmune disease. The possibility that a given inhibitory receptor may have opposing effects on the generation of autoreactivity at different checkpoints in different contexts may impact upon therapy targeted at inhibitory receptors, helping explain the mechanism of new therapeutic agents (e.g. anti-CD22 therapy epratuzumab, which generated unexpected dose-related complexity in clinical studies[63]), but perhaps also creating opportunities for synergistic therapies targeting individual checkpoints.

## Methods

**Mice**. The SW$_{HEL}$ mice were a kind gift of Robert Brink. They were crossed with the FcγRIIb-deficient mouse strain[35], the FcγRIIb-BTG strain (itself crossed to the CD45.1 background)[40] and the FcγRIIb$^{wild/H1}$ KI mouse strain[44]. FcγRIIb KO mice (C57BL/6 FcγRIIb$^{-/-}$) were provided by J. Ravetch and S. Bolland. This strain was originally generated on the 129/sv background and backcrossed for more than 15 generations on the C57BL/6 background. Despite this extensive back-crossing, we cannot exclude the persistence of 129/sv-derived polymorphisms in the *Fcgr2b* genetic region. Nevertheless, as we also used the FcγRIIb-BTG and the FcγRIIb$^{wild/H1}$ KI strains that were both generated on pure C56BL/6 background, we are confident that the results observed are indeed due to the level of expression of FcγRIIb. *Fcgr2b* is on the same chromosome than the *Ly5* locus, making it very difficult to obtain FcγRIIb KO mice bearing the congenic marker CD45.1. The mHEL (membrane-bound HEL transgenic) and ML5 (soluble HEL transgenic) mice have been described previously[2,5]. We crossed the mHEL strain with the *Rag2$^{-/-}$* (here called RAG2KO) strain to obtain the mHEL × RAG2KO strain, expressing one allele of the mHEL transgene. For all experiments, the number of animals included in each experiments was determined based on our previous expertise using mouse models with different expression level of FcγRIIb and by using the G*Power software for *a priori* power calculation.

To generate bone marrow chimeras, recipient mice were irradiated lethally (1000 Gy for WT, mHEL and mL5 recipients) or sub-lethally (500 Gy for RAG2KO and mHEL × RAG2KO recipients) and reconstituted by intravenous injection of fresh bone marrow cells. Reconstitution was assessed 6 weeks later and mice that were not reconstituted fully were excluded. Reconstituted mice were randomly numbered independently of their genotype or BM used for their reconstitution and analysed at least 8 weeks post irradiation. Mice were immunised with sheep red blood cells (SRBC) conjugated with HEL or SRBC alone by intra-peritoneal injection, as indicated[64]. All experiments were performed according to the regulations of the UK Home Office Scientific Procedures Act (1986) and permission was given by the UK Home Office to perform such procedures at our local animal facility.

**Flow cytometry**. Single cell suspensions of the spleen and bone marrow were prepared by mashing or flushing the organs, respectively[44]. For phosphoflow staining, splenocytes were stimulated at 37 °C for 5 min with a F(ab')2 goat anti-mouse IgG (H + L) antibody (from Jackson immunosearch, 10 μg/ml). Cells were then fixed with ice-cold methanol on ice for 30 min before staining with the relevant antibodies. HEL-specific B cells were detected by staining with either recombinant HEL (40 ng/ml, Sigma-Aldrich) followed by a rabbit anti-HEL biotinylated (AbCam) and streptavidin eFluor450 (eBioscience) or 1 μg/ml of biotinylated HEL (conjugation with the EZ-link Sulfo-NHS-biotin kit from Pierce following the manufacturer's instructions) and streptavidin eFluor450 (eBioscience). For all experiments, the gating of HEL-binding cells was set up based on control staining with a biotinylated rabbit anti-HEL and streptavidin eFluor450, or streptavidin eFluor450 alone when biotinylated HEL was

used. For the analysis, the HEL+ gates were drawn on the condition minus HEL and applied to all other samples.

FACS analysis was performed on a BD LSR Fortessa (BD biosciences) and data were analysed with the Flowjo software (TreeStar, Ashland, OR). All antibodies used are described in Supplementary Table 2.

**ELISA and ELISPOT**. Serum HEL-specific IgG1 were detected by ELISA using a modified version of the protocol previously described[44]. Briefly, plates were coated with 5 μg/ml of HEL in PBS overnight at 4 °C. Sera were added to the saturated plates and were incubated for 2 h at 37 °C. Pooled sera from hyperimmune mice (immunised with SRBC-HEL twice) were used as a standard. Plates were developed with the BD Opteia kit (BD Biosciences). Serum dsDNA-specific IgG1 was detected by ELISA as previously described[65].

For detection of HEL-specific antibody-forming cells (AFCs) ELISPOT plates (Millipore) were coated with 5 μg/ml of HEL in PBS overnight at 4 °C. Single-cell suspensions of the spleen were added to saturated ELISPOT plates in quadruplicate and incubated overnight at 37 °C in 5% CO$_2$ in a humidified incubator with culture medium. AFCs were detected with goat anti-mouse IgG1 antibody conjugated to horseradish peroxidase (Southern Biotech). Plates were developed using 3-amino-9-ethylcarbazole tablets (Sigma-Aldrich). Plates were read using an AID ELISpot reader according to the manufacturer's instructions.

**In vitro proliferation and differentiation**. Splenocytes were labelled with 50 nM CFSE for 10 min before being washed twice in cold culture medium (RPMI, 10% FCS, penicillin/streptomycin, sodium pyruvate). Cells were then plated at $2 \times 10^6$/ml in round bottom 96-well plates and stimulated for 96 h with 10 μg/ml of *E. coli* lipopolysaccharide (LPS) (Sigma-Aldrich). B-cell proliferation was measured by quantifying CFSE dilution by flow cytometry. B-cell differentiation was measured by quantifying the percentage of CD138$^{hi}$B220$^{lo}$ plasmablasts by flow cytometry.

**Statistical analysis of the mouse data**. When three groups were compared, the *p*-values were determined with the Prism software using the Kruskal–Wallis non-parametric test (significant *p*-values are indicated on the figures). When two groups were compared, the *p*-values were determined with the Prism software using the Mann–Whitney unpaired two-tailed non-parametric test (significant *p*-values are indicated on the figures by stars; *$p < 0.05$, **$p < 0.01$, ***$p < 0.001$, ****$p < 0.0001$). Non-significant *p*-values are indicated by "n.s".

**Cell sorting for the human repertoire analysis**. Peripheral blood mononuclear cells (PBMCs) were obtained from healthy individuals homozygous for the FcγRIIB-T232 or FcγRIIB-I232 site, under appropriate ethics approval from the NIHR Cambridge Bioresource. Inclusion criteria for individuals were people aged between 44 and 77 years, with no serious co-morbidities, no direct family history of autoimmune disease, no use of immunosuppressants or steroids and no hospitalisation within the last 12 months. Individuals were age and sex matched (18 females and 11 males matched between genotypes). Flow sorting was performed using CD19-BV785, CD38-BV711, CD3-NC650, CD14-605NC, CD24-PerCP-Cy5.5, IgD-FITC, CD27-PE-Cy7 (all from BD Bioscience) and Aqua (for live-dead cell detection, Invitrogen), where flow protocol is outlined in Supplementary Fig. 7.

**Inclusion criteria for the SLE cohort**. Inclusion criteria for healthy individuals were people aged between 20 and 77 years, with no serious co-morbidities, no direct family history of autoimmune disease, no use of immunosuppressants or steroids, and no hospitalisation within the last 12 months.

The SLE cohort comprised patients attending or referred to the Addenbrooke's Hospital specialist vasculitis unit between July 2004 and June 2016 who met at least four American College of Rheumatology SLE criteria[66], presenting with active disease (defined as a) new BILAG score of A or B in any system (including switch from C to A or D to B), b) physician assessment of active disease, and c) a requirement for an increase of immunosuppressive treatment (excluding isolated increase/commencement of prednisolone at a dose ≤25 mg). After treatment with an immunosuppressant, patients were followed up monthly. Disease monitoring was undertaken with serial British Isles Lupus Assessment Group disease scoring (BILAG, a major international SLE activity scoring system)[67] and serum anti-nuclear antibody status.

Ethical approval for this study was obtained from the Cambridge Local Research Ethics Committee (reference numbers 04/023, 08/H0306/21, 08/H0308/176) and Eastern NHS Multi Research Ethics Committee (07/MRE05/44), with informed consent obtained from all subjects enrolled.

**Reverse transcription and amplification**. Repertoire analysis was performed as described[52]. Briefly, RNA extraction was performed using RNeasy Micro Kit (Qiagen) according to the manufacturer's protocol. Reverse transcription (RT) was performed in a 23-μL reaction with SuperScript®III (Thermo Fisher), RNA template, 12 μM reverse primer, nuclease-free water and incubated for 5 min at 70 °C. This mixture was immediately transferred to ice for 1 min, and the RT mix 2 (4 μL 5x FS buffer, 1 μL DTT (0.1 M), 1 μL dNTP (10 mM), 1 μL SuperscriptIII) was added and incubated at 50 °C for 60 min followed by 15 min inactivation at 70 °C.

cDNA was cleaned-up with Agencourt AMPure XP beads and PCR amplified with V-gene multiplex primer mix (10 μM each forward primer) and 3′ universal reverse primer (10 μM) using KAPA protocol and the thermal cycling conditions: 1 cycle (95 °C—5 min); 5 cycles (98 °C—5 s; 72 °C—2 min); 5 cycles (65 °C—10 s, 72 °C—2 min); 25 cycles (98 °C—20 s, 60 °C— 1 min, 72 °C—2 min) and 1 step (72 °C—10 min). Primers are provided in Supplementary Table 3.

**IgH sequencing and analysis.** MiSeq libraries were prepared using Illumina protocols and sequenced using 300 bp paired-ended MiSeq (Illumina). Raw MiSeq reads were filtered for base quality (median Phred score > 32) using QUASR (http://sourceforge.net/projects/quasr/)[68]. MiSeq forward and reverse reads were merged together if they contained identical overlapping regions of >50 bp, or otherwise discarded. Universal barcoded regions were identified in reads and orientated to read from V-primer to constant region primer. The barcoded region within each primer was identified and checked for conserved bases (i.e. the T's in NNNNTNNNNTNNNNT). Primers and constant regions were trimmed from each sequence, and sequences were retained only if there was >80% sequence certainty between all sequences obtained with the same barcode, otherwise discarded. The constant region allele with highest sequence similarity was identified by 10-mer matching to the reference constant region genes from the IMGT database[69], and sequences were trimmed to give only the region of the sequence corresponding to the variable (V–D–J) regions. Isotype usage information for each IgH was retained throughout the analysis hereafter. Sequences without complete reading frames and non-immunoglobulin sequences were removed and only reads with significant similarity to reference IgHV and J genes from the IMGT database were retained using BLAST[70]. IgH gene usages and sequence annotation was performed in IMGT V-QUEST, where repertoire differences were performed by custom scripts in python, and statistics were performed in R using Wilcoxon tests for significance (non-parametric test of differences between distributions).

**Software and algorithms for repertoire analysis.** The following software and algorithms were used: QUASR[68] (http://sourceforge.net/projects/quasr/), BLAST[70] (https://blast.ncbi.nlm.nih.gov/Blast.cgi), IMGT V-QUEST[69] (http://www.imgt.org/HighV-QUEST/), Immune_receptor_NETWORK-GENERATION[71] (https://github.com/rbr1/Immune_receptor_NETWORK-GENERATION) and R version 3.3.3 (2017-03-06, https://www.r-project.org).

**Reporting Summary.** Further information on experimental design is available in the Nature Research Reporting Summary linked to this article.

## Data availability

A reporting summary for this article is available as a Supplementary Information file. The source data underlying Figs. 1 to 6 and Supplementary Figs. 1–6 are available as a Source Data file. EGA accession numbers for sequencing data are provided in Supplementary Table 4 (https://www.ebi.ac.uk/ega/). All other relevant data are available from the authors upon reasonable requests.

## Code availability

IGH sequence analysis was performed using Immune_receptor_NETWORK-GENERATION (https://github.com/rbr1/Immune_receptor_NETWORK-GENERATION).

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

## Acknowledgements

We are grateful to Robert Brink for providing the SW_HEL mice. We thank Devina Devikar, Alexander Hatton, Alessandra De Riva, James Lee and Tim Wilson for technical help. We gratefully acknowledge Valerie Morrison, all National Institute for Health Research (NIHR) Cambridge BioResource volunteers, the NIHR Cambridge BioResource centre and staff and NHS Blood and Transplant. This research was supported by the Cambridge NIHR BRC Cell Phenotyping Hub. In particular, we wish to thank Anna Petrunkina Harrison, Chris Bowman, Natalia Savinykh and Esther Perez for their advice and support in flow cytometry. This work was funded by the Wellcome Trust (Programme Grant Number 083650/Z/07/Z and Wellcome Trust Investigator Award 200871/Z/16/Z to KGCS and Sir Henry Wellcome Fellowship WT106068AIA to RBG) and supported by the NIHR Cambridge Biomedical Research Centre. ME was funded by the Wellcome Trust (Programme Grant Number 083650/Z/07/Z), by a Junior Team Leader starting grant from the Laboratory of Excellence in Research on Medication and Innovative Therapeutics (LabEx LERMIT) supported by a grant from ANR (ANR-10-LABX-33) under the programme "Investissements d'Avenir" (ANR-11-IDEX-0003-01) and by an ANR @RAction starting grant (ANR-14-ACHN-0008). KGCS is an NIHR Senior Clinical Investigator and a Distinguished Innovator of the Lupus Research Institute.

## Author contributions

M.E. and K.G.C.S. designed the project. M.E. performed the experiments and analysed the data. R.B.R. performed the human analysis. J.M.S., N.A., L.W. and A.E.D. assisted with experiments, M.A.L. with project design and K.G.C.S. with the data analysis. M.E. and K.G.C.S. wrote the paper with input from all other authors.

## Additional information

**Competing interests:** The authors declare no competing interests.

