## [Peer Review File · Nature Communications]

Reviewers' comments:

Reviewer #1 (Autoimmune, auto Ab, Ab repertoire)(Remarks to the Author):

The manuscript by Espeli et al attempts to clarify the role of the inhibitory IgG receptor FcγRIIb in B cell tolerance. The work presented is based primarily on mouse models, supplemented with some human data including volunteers and patients bearing a particular SNP in the *fcgr2b* gene. The experimental setup for mouse experiments consists essentially of bone marrow transfer of donor mice expressing an anti-HEL antibody heavy chain (SW-HEL) mice in lethally irradiated wt or HEL-transgenic (membrane-bound or soluble) recipient mice. SW-HEL donor mice are wt, FcγRIIb-KO or expressing FcγRIIb only on B cells and at higher levels than in wt mice (BTG mice; expression 2 to 10-fold higher FcγRIIb on B cell subsets than wt mice). B cell numbers and phenotypes are compared in recipient mice.

Although this manuscript represents a substantial amount of work, the differences observed between donor mice genotypes in most experiments are rarely statistically different. Statistical differences appear mainly when comparing FcγRIIb-KO with BTG donor mice (Figs 1 & 3; most comparisons of Figs 4 & 5) that represent a very extreme situation that may not find any physiological relevance. Many general conclusions are based on this extreme comparison and rarely on FcγRIIb-KO versus wt dataset comparisons that should be producing compelling results if more mice would have been used per group. This severely weakens the conclusions of this manuscript. Also problematic, general conclusions of each paragraph do not integrate the lack of significance of comparisons with the wt group.

The human data provided is convincing.

The proposition that FcγRIIb has opposite effects on tolerance (pre- versus post-immune) and that FcγRIIb controls bystander (autoreactive) B cell activation is however very interesting, but suffers from the following:

Major concerns:

1) Most comparisons that would have physiological relevance are not statistically significant. Authors should use larger cohorts of mice or pool data from repeat experiments to reach significance and support their claims.

2) Conclusions need to be carefully re-written to avoid overstatements based on generalization from significant and non-significant comparisons. Examples on lines:

121 – in fact comparing FcγRIIb absence on all cells versus overexpression on B cells, and not “reduced expression”

130,131: n.s. comparisons to wt for T1, mature and MZ cells

199,200 and 205: conclusions only apply to comparisons with BTG but not wt

230,231: all comparisons in RAG2^{-/-} mice datasets (Fig. 6c,d,e) are n.s.

238-241: mix of references to n.s. data and statistically significant data

3) Authors should avoid the terminology “slight increase” (line 178, 202) or “slight decrease” when describing non-statistically different comparisons. Reply to major concern #1 should solve this point.

4) The discussion should integrate the limitations of the models used, and in particular how FcγRIIb absence on all cells versus expression only on B cells subsets at a 2-10 fold level (Fig1C of ref 40) allows conclusions to be drawn.

5) Fig.7 should include the same analysis “per subset” as provided in 7b also for data from SLE patients to enable comparisons between T232 healthy and T232 SLE patients. The comparison made

on line 267 does not seem appropriate.

Minor concerns:

- 1) line 135: the concept of "tonic" signaling should be explained or properly referenced
- 2) Fig.4b: explain or show how HEL+ and HEL- gates have been set
- 3) Fig.S6b: a p value of 0.086 is non-significant and should be indicated as such. Authors should opt for n.s. in all figures or numerical p values in all figures to avoid confusion.

Reviewer #2 (BCR, repertoire, B response)(Remarks to the Author):

This is an interesting investigation of the association between expression of Fc γ RIIb and B cell checkpoint failure. Since genetic variations affecting the expression of Fc γ RIIb are associated with development of lupus, this is important.

Using a HEL system with mice that have reduced, wild type or increased expression of Fc γ RIIb the authors show that checkpoints in the bone marrow or as B cells mature in the periphery or anergy are not affected by reduced expression of Fc γ RIIb. If anything, selection against autoreactivity is more stringent when Fc γ RIIb is lacking. The authors show however that autoimmunity in absence of Fc γ RIIb can emerge from the germinal center response. The authors make a good case in their discussion for the model they use and argue that their observations are due to B cell intrinsic properties, though for most of the work this is not assured.

The negative data in figures 1-3 seem clear though it would be better to see data as dot plots for transparency. Also, since 2 different statistical tests are used, their outcomes could be color coded (possibly different colored bars, stars or values depending on the test) so the reader can know more easily which tests are providing this information.

My major concern with the manuscript is that the mouse and human parts are not merged well in the manuscript to be mutually supportive. Possible important similarities and differences between the mouse and human data are not brought together or well discussed.

The study of mice detects an increase in autoreactivity in the cells generated in germinal centers and an increase in GC cells per se. Measurements include titres of IgG anti-DNA antibodies that are enhanced in mice with reduced functional Fc γ RIIb immunised with HEL-SRBC. In contrast to the class switched cells and antibodies identified as autoreactive in the studies of mice, the analysis of human B cells homozygous for the lupus predisposing allele of Fc γ RIIb referred to as T232 identifies a higher frequency of IGHV4-34 in the CD27+IgD+ population (that expresses IgM), compared to controls. Many consider the CD27+IgD+ population to be marginal zone B cells and not memory cells (eg. Descatoire et al, JEM 2014 <https://www.ncbi.nlm.nih.gov/pubmed/24733829>). In contrast there was no reported increase in IGHV4-34 indicating lack of checkpoint selection in the CD27+IgD- population that would include class switched or unswitched memory cells.

A higher frequency of IGHV4-34 was also observed in B cells referred to as 'activated' but it is unclear to this reviewer what these cells are. The definition of 'activated' is confusing and unfamiliar. The text mentions CD29+. However, the gating figure S7 has CD38 as the relevant parameter for definition of activated cells with no mention of CD29. In figure S7 activated cells appear to be derived from the IgD+CD27- quadrant which agrees with the text, but naïve cells come from a CD27-IgD- quadrant, that must be a mistake because naïve B cells express IgD. I think this figure S7 is possibly wrongly

drawn and in any case disagrees with the text that mentions CD29. In figure S9, Naive B cells that are widely considered to be the largest B cell subset in blood appears too small. Could naive B cells be in the IgD+CD27- figure subpart? Figure S10 introduces pre/early GC. How were these gated in blood? To avoid further confusion the gating of cells should be illustrated as actual plots from an experiment rather than diagrams. CD27+IgD+ cells and other subsets should be referred to by phenotype, eg CD27+IgD+ rather than names such as 'IgD+ memory' or 'activated' to ensure accuracy and lack of ambiguity in named subset assignment.

It is also It is not clear if data from the 'activated' subset comprises multiple subsets or if all isotypes have equivalent use of IGHV4-34 (or only M?).

To this reviewer, the human data as presented suggests that selection of CD27+IgM+IgD+ (CD27+IgD+ from sorting IgM from sequence analysis) is affected by lack of functional FcγRIIb. Class switched memory appears unaffected and therefore the parallels with mice are not as sharp as the manuscript implies, though species difference may contribute to this.

Reviewer #3 (Germinal center response, autoimmune)(Remarks to the Author):

The authors demonstrate interesting opposing effects of FcγRIIb on B cell selection. While FcγRIIb has a well-known inhibitory role during antigen-dependent B cell activation and in that way inhibits inappropriate B cell activation, during central tolerance induction it seems to have the opposite effect. This shows during development in fewer autoreactive HEL-specific self-reactive B cells being selected, if membrane expressed HEL autoantigen is present and FcγRIIb is absent, and more self-reactive B cells if FcγRIIb is overexpressed. The fewer B cells developing in absence of FcγRIIb are anergic. During antigen-dependent activation, however, in the absence of FcγRIIb, more self-reactive B cells seem to enter GC responses. Similar results are found when weaker tolerance induction is studied (hosts expressing soluble HEL instead of membrane expressed HEL). It seems that in the absence of FcγRIIb non-antigen-specific autoreactive B cells are allowed to enter GC responses. V-gene repertoire expression data from humans support this.

It is concluded that FcγRIIb has opposing roles during preimmune repertoire selection and during antigen-induced B cell differentiation. This makes sense: In the preimmune phase FcγRIIb inhibits BCR signalling, therefore counteracts tolerance induction induced by inappropriate BCR signals. In the antigen-dependent phase of B cell differentiation, FcγRIIb also inhibits BCR signalling. Now, paradoxically, FcγRIIb leads to better control of autoreactive B cells.

The authors make the point that FcγRIIb inhibits recruitment of non-specific "bystander" cells into the GC response. I am having problems with that definition. It is difficult to measure whether GC B cells are really non-specific or just very low affinity. Therefore, there is no way of knowing whether what is seen is real non-specific bystander activation or just a broader activation of cross-reactive B cells.

The data presented on antigen-dependent activate are somewhat incomplete: In Fig. 4 and 5 cells entering GC responses are only given as percentages (HEL-specific cells per GC cells), but not in absolute numbers of HEL-specific cells and no size of the GC response is shown. Therefore, it is not clear whether the number of antigen-specific cells recruited into GCs is high or low and how absolute numbers compare between groups. Further, the upper half of Fig. 4b shows only 3 plots each. Are these from wt or mHEL hosts? And how do wt or mHEL hosts compare? Also, Fig. 4e is problematic.

Dividing percentages does not make much sense, if the size of the parent populations is not known. Dividing absolute numbers would be better.

In Fig. 6, however, we are told absolute numbers of antigen-specific cells without being shown relative percentages. Therefore, again the reader cannot conclude whether the interpretation of the authors – that fewer non-specific B cells enter GCs in the absence of FcRIIb – is true or whether just total GC sizes are changing.

Minor point:

Fig. 4F: what does the axis label mean ("5.105 cells")?

Point-by-point response to reviewers' comments:

We are grateful for the thorough evaluation and constructive comments provided by the three reviewers - these have helped us to significantly improve the manuscript. In particular, we have pooled data to achieve more reliable statistical conclusions, and we have standardized and homogenized data presentation throughout the manuscript. Moreover, we have corrected some typographic errors and performed a more comprehensive analysis in the human part of the paper which has clarified its message, and allowed more informative discussion of the discrepancies and similarities between human and mouse data. We have amended the figures and text accordingly, and specific comments are addressed below.

Reviewer #1 (Autoimmune, auto Ab, Ab repertoire)(Remarks to the Author):

The manuscript by Espeli et al attempts to clarify the role of the inhibitory IgG receptor FcγRIIb in B cell tolerance. The work presented is based primarily on mouse models, supplemented with some human data including volunteers and patients bearing a particular SNP in the fcgr2b gene. The experimental setup for mouse experiments consists essentially of bone marrow transfer of donor mice expressing an anti-HEL antibody heavy chain (SW-HEL) mice in lethally irradiated wt or HEL-transgenic (membrane-bound or soluble) recipient mice. SW-HEL donor mice are wt, FcγRIIb-KO or expressing FcγRIIb only on B cells and at higher levels than in wt mice (BTG mice; expression 2 to 10-fold higher FcγRIIb on B cell subsets than wt mice). B cell numbers and phenotypes are compared in recipient mice.

Although this manuscript represents a substantial amount of work, the differences observed between donor mice genotypes in most experiments are rarely statistically different. Statistical differences appear mainly when comparing FcγRIIb-KO with BTG donor mice (Figs 1 & 3; most comparisons of Figs 4 & 5) that represent a very extreme situation that may not find any physiological relevance.

Many general conclusions are based on this extreme comparison and rarely on FcγRIIb-KO versus wt dataset comparisons that should be producing compelling results if more mice would have been used per group. This severely weakens the conclusions of this manuscript. Also problematic, general conclusions of each paragraph do not integrate the lack of significance of comparisons with the wt group.

The human data provided is convincing.

The proposition that FcγRIIb has opposite effects on tolerance (pre- versus post-immune) and that FcγRIIb controls bystander (autoreactive) B cell activation is however very interesting, but suffers from the following:

Major concerns:

1) Most comparisons that would have physiological relevance are not statistically significant. Authors should use larger cohorts of mice or pool data from repeat experiments to reach significance and support their claims.

We thank the reviewer for this helpful comment. We have now pooled data from different experiments as suggested, which has greatly improved the statistical confidence in the conclusions. In most cases comparison of FcγRIIb KO and WT animals now reaches statistical significance. The figures and text have been changed accordingly.

2) Conclusions need to be carefully re-written to avoid overstatements based on generalization from significant and non-significant comparisons. Examples on lines:

121 - in fact comparing FcγRIIb absence on all cells versus overexpression on B cells, and not "reduced expression".

With pooled data significant differences were observed between KO and WT groups as well as between WT and BTG groups, and thus the original text is now accurate.

130,131: n.s. comparisons to wt for T1, mature and MZ cells.

Based on the new analyses the sentence was rewritten as follow: "Supporting this observation, B cell-specific overexpression of FcγRIIb was associated with increased frequency of HEL-specific B cells compared to KO, and, in some populations WT, groups, consistent with reduced tolerance at the transitional T2/T3 (Fig. 2d) and mature follicular (Fig. 2e-f) B cell stages."

199,200 and 205: conclusions only apply to comparisons with BTG but not wt

The sentence has been modified as follow for clarity: “Following immunization, we observed an increase in the frequency and number of HEL-specific GC B cells (Fig. 5e) and in the ratio of HEL-specific GC to follicular B cells (Fig. 5f) in sHEL recipient mice reconstituted with SW_{HEL}-FcγRIIb-KO BM compared to mice reconstituted with SW_{HEL}-FcγRIIb-BTG BM”

230,231: all comparisons in RAG2^{-/-} mice datasets (Fig. 6c,d,e) are n.s.

The sentence has been modified as follow for clarity: “As expected, enhanced pre-immune tolerance resulted in fewer HEL-specific B cells in mHEL recipient mice reconstituted with SW_{HEL}-FcγRIIb-KO compared to SW_{HEL}-FcγRIIb-WT and SW_{HEL}-FcγRIIb-BTG BM cells (Fig. 6c). However SRBC immunization drove an increase in HEL-specific GC B cells (Fig. 6d) and in serum HEL-specific IgG1 (Fig. 6e) in RAG2^{-/-} mHEL mice reconstituted with SW_{HEL}-FcγRIIb-KO BM compared to SW_{HEL}-FcγRIIb-BTG BM cells.”

238-241: mix of references to n.s. data and statistically significant data

These sentences have been modified accordingly: “There was enhanced tolerance at the pre-immune checkpoints (reduced frequency of HEL-specific autoreactive B cells in BM and spleen of SW_{HEL}-FcγRIIb^{wild/H1} KI/mHEL mice compared to control: Fig. S6b and c). Despite this, in the SW_{HEL}-FcγRIIb^{wild/H1} KI chimeric mice immunized with SRBC, the total number of HEL-specific GC B cells as well as the ratio of HEL-specific GC B cells to total splenic B cells were increased (Fig. S6 d-e). A trend towards increased serum HEL-specific autoantibodies in mice reconstituted with SW_{HEL}-FcγRIIb^{wild/H1} KI BM was also observed (Fig. S6f).”

Other conclusions have been modified in light of the new statistical analysis of pooled data – these are highlighted in red.

3) Authors should avoid the terminology “slight increase”; (line 178, 202) or “slight decrease”; when describing non-statistically different comparisons. Reply to major concern #1 should solve this point.

We agree with the reviewer that this terminology was potentially misleading. Reanalysis of pooled data has resulted in both of these differences now being statistically significant, and so “slight” has been removed from the text.

4) The discussion should integrate the limitations of the models used, and in particular how FcγRIIb absence on all cells versus expression only on B cells subsets at a 2-10 fold level (Fig1C of ref 40) allows conclusions to be drawn.

The discussion has now been modified accordingly (p13), to include a paragraph on the limitations of the models used, and the way in which the use of complementary model systems overcomes that:

“Three distinct mouse models were used in this study, as each alone had inherent limitations. The FcγRIIb-deficient mouse has a profound defect affecting not only B cells, but also other cells implicated in the immune response (Bolland et al. 2000, Boross et al. 2011). We therefore also used two additional models. In one FcγRIIb was over-expressed on all B cells by between 2 and 10-fold (but was expressed at normal levels on other cell types (Brownlie et al. 2008). The other is a more “physiological” model of FcγRIIb under-expression, in which a “knock-in” of a naturally occurring promoter variant results in a failure of activation-induced upregulation of FcγRIIb specifically on B cells (Espéli et al. 2012). This genetic model would be expected to give a less extreme phenotype than the deficient mice, as its defect is subtle, but would confirm the B cell-specificity of any effect. Taken together comparisons between these three models increase confidence in our findings, as they provide confirmation in two underexpression models – one an “extreme” knockout, and one a subtle B-cell specific “physiological” genetically modified mouse – and a cell-specific overexpression model.”

5) Fig.7 should include the same analysis “per subset” as provided in 7b also for data from SLE patients to enable comparisons between T232 healthy and T232 SLE patients. The comparison made on line 267 does not seem appropriate.

The repertoire analysis on SLE patients was performed on PBMCs from a unique cohort and we do not have access to samples to perform such an analysis on sorted cell subsets. However, to investigate whether this enhanced usage of *IGHVH4-34* in SLE patients is driven by pre- or post-immune B cell subsets we performed a computational segregation of our sequence data. In brief, sequences were segregated in 3 groups on the basis of the isotype (IgD/IgM or class switched) and of the mutational status (presence of somatic hyper mutation or

not). IgD/IgM unswitched sequences should correspond mainly to naïve B cells while mutated and class-switched sequences should mainly correspond to antigen-experienced B cells. As shown in the new figure 7a, and noted in the text on p12, *IGHV4-34* was enriched among SLE sequences without evidence of switching or somatic hyper mutation (predominantly naïve B cells), as well as sequences corresponding to antigen-experienced B cells, as evidenced by isotype switching or somatic hyper mutation, suggesting a defect in both pre- and post-immune tolerance in SLE.

Minor concerns:

1) line 135: the concept of “tonic” signaling should be explained or properly referenced

The sentence has been modified for improved clarity and references have been added (Fournier *et al.* J Immunol 2008; Myers *et al.* Trends in Immunology 2017).

2) Fig.4b: explain or show how HEL+ and HEL- gates have been set

As mentioned in the material and methods (page 18), for this panel as for all experiments presented in this manuscript the gating of HEL-binding cells was set up based on control staining without HEL but with streptavidin eFluor450. Methods have been expanded to better explain our gating strategy.

3) Fig.S6b: a p value of 0.086 in non-significant and should be indicated as such. Authors should opt for n.s. in all figures or numerical p values in all figures to avoid confusion.

Non-significant p-values are now indicated by n.s. in all figures.

Reviewer #2 (BCR, repertoire, B response)(Remarks to the Author):

This is an interesting investigation of the association between expression of FcγRIIb and B cell checkpoint failure. Since genetic variations affecting the expression of FcγRIIb are associated with development of lupus, this is important.

Using a HEL system with mice that have reduced, wild type or increased expression of FcγRIIb the authors show that checkpoints in the bone marrow or as B cells mature in the periphery or anergy are not affected by reduced expression of FcγRIIb. If anything, selection against autoreactivity is more stringent when FcγRIIb is lacking. The authors show however that autoimmunity in absence of FcγRIIb can emerge from the germinal center response. The authors make a good case in their discussion for the model they use and argue that their observations are due to B cell intrinsic properties, though for most of the work this is not assured.

We now discuss B cell specificity in the new paragraph in the discussion outlining the limitations of the strains used (added in response to Reviewer 1 – see above and page 13).

The negative data in figures 1-3 seem clear though it would be better to see data as dot plots for transparency. Also, since 2 different statistical tests are used, their outcomes could be color coded (possibly different colored bars, stars or values depending on the test) so the reader can know more easily which tests are providing this information.

We now indicate in all panels whether the p-value is significant or not and, as suggested by the Reviewer, have color-coded this information to allow immediate identification of the test. Mann-Whitney non-parametric tests are indicated with black stars, while p-values obtained when comparing three groups with the non-parametric Kruskal-Wallis test are indicated in purple (labeled “KW”). Finally, raw data for all panels are available as Source Data file. All relevant figures and legends have been altered to incorporate these changes.

My major concern with the manuscript is that the mouse and human parts are not merged well in the manuscript to be mutually supportive. Possible important similarities and differences between the mouse and human data are not brought together or well discussed.

This is a fair point, and we have modified the discussion to include a more nuanced and comprehensive comparative analysis of the mouse and human data. This is addressed in more detail below.

The study of mice detects an increase in autoreactivity in the cells generated in germinal centers and an increase in GC cells per se. Measurements include titres of IgG anti-DNA antibodies that are enhanced in mice with reduced functional FcγRIIb immunised with HEL-SRBC. In contrast to the class switched cells and antibodies identified as autoreactive in the studies of mice, the analysis of human B cells homozygous for the lupus predisposing allele of FcγRIIb referred to as T232 identifies a higher frequency of IGHV4-34 in the CD27+IgD+ population (that expresses IgM), compared to controls.

Many consider the CD27+IgD+ population to be marginal zone B cells and not memory cells (eg. Descatoire et al, JEM 2014 <https://www.ncbi.nlm.nih.gov/pubmed/24733829>).

The reviewer is correct – we have described this population inaccurately. As pointed out, the CD27+IgD+ population is most likely MZ derived (Descatoire et al, JEM 2014 Bagnara et al., 2015), and perhaps also contains human B1 B cells (Krutzmann et al., 2003) and memory B cells that exited GCs prior to isotype switching (Tangye and Good, 2007). The level of somatic hypermutation observed in our sequencing data for this B cell subset is consistent with antigen-experience (mean mutations in this subset: 9.7bp per IgH). We have therefore reannotated the text (e.g. in line 268-269) and figures to reflect this B cell phenotype :
“We flow-sorted peripheral blood CD19⁺IgD⁺CD27⁻ naive/T1/2/3 B-cells; CD19⁺IgD⁻CD27⁺CD38^{mid/hi} IgM and switched activated B-cells; CD19⁺IgD⁺CD27⁺IgD⁺ B cells that include marginal zone B cells, and perhaps “B1-like” and unswitched memory cells (48-51); CD19⁺IgD⁻CD27⁺CD38⁻ switched memory B-cells; and CD19⁺IgD⁻CD27⁺CD38⁺ plasmablasts (Fig. S7).”

In contrast there was no reported increase in IGHV4-34 indicating lack of checkpoint selection in the CD27+IgD- population that would include class switched or unswitched memory cells.

A higher frequency of IGHV4-34 was also observed in B cells referred to as “activated” but it is unclear to this reviewer what these cells are. The definition of 'activated' is confusing and unfamiliar. The text mentions CD29+. However, the gating figure S7 has CD38 as the relevant parameter for definition of activated cells with no mention of CD29. In figure S7 activated cells appear to be derived from the IgD+CD27- quadrant which agrees with the text, but naive cells come from a CD27-IgD- quadrant, that must be a mistake because naive B cells express IgD. I think this figure S7 is possibly wrongly drawn and in any case disagrees with the text that mentions CD29.

In figure S9, Naive B cells that are widely considered to be the largest B cell subset in blood appears too small. Could naive B cells be in the IgD+CD27- figure subpart? Figure S10 introduces pre/early GC. How were these gated in blood? To avoid further confusion the gating of cells should be illustrated as actual plots from an experiment rather than diagrams. CD27+IgD+ cells and other subsets should be referred to by phenotype, eg CD27+IgD+ rather than names such as “IgD+ memory” or “activated” to ensure accuracy and lack of ambiguity in named subset assignment.

The reviewer has been understandably misled by some key errors, for which we apologize and which we have now corrected. They include:

- “CD29” should read “CD19” in the text
- Figure S7 has been corrected, restoring the naïve B cell population to its correct size.
- Cells that were described as “activated” (and, confusingly, also as “pre/early GC” – a term we have now removed entirely) were CD19+IgD-CD27-CD38^{mid/hi}. This population has the highest level of IgG3, the first isotype to which a B cell can switch, supporting recent activation, but, as suggested by the reviewer, to remove potential ambiguity the subsets will henceforth be described by their phenotype.
- Actual plots are now provided for Figure S7.

It is also It is not clear if data from the “activated” subset comprises multiple subsets or if all isotypes have equivalent use of IGHV4-34 (or only M?).

The reviewer is correct in surmising that CD19+IgD-CD27-CD38^{mid/hi} subset comprised a mixed population of B cells, some are IgM+, and others have undergone class-switching (see Figure S10 for details). The IGHV4-34 usage was higher in the T232 individuals and this was more marked in the IgM+ B cells.

To this reviewer, the human data as presented suggests that selection of CD27+IgM+IgD+ (CD27+IgD+ from sorting IgM from sequence analysis) is affected by lack of functional FcγRIIb. Class switched memory appears unaffected and therefore the parallels with mice are not as sharp as the manuscript implies, though species difference may contribute to this.

We agree with this interpretation, and have now amended the Results section (p12):

“IGHV4-34 was enriched among BCRs without sequence evidence of switching or SHM (comprising predominantly naïve B cells), as well as antigen-experienced BCRs as evidenced by isotype switching or SHM, suggesting a defect in both pre- and post-immune tolerance (Fig. 7a).

In I232 healthy volunteers, the frequency of IGHV4-34 gene usage was higher in CD19⁺IgD⁺CD27⁻ than in CD19⁺IgD⁻CD27⁻CD38^{mid/hi} and CD19⁺CD27⁺IgD⁻ populations (Fig. 7b), consistent with the impact of peripheral tolerance on this autoreactive population. Healthy T232 and I232 individuals had similar IGHV4-34 proportions in CD19⁺IgD⁺CD27⁻ cells, suggesting the enhanced central tolerance seen in FcγRIIB-deficient mice may not occur in humans. IGHV4-34 was, however, enriched in CD19⁺CD27⁺IgD⁺ and CD19⁺IgD⁻CD27⁻CD38^{mid/hi} populations to a level similar to that seen in SLE patients (Fig. 7b), and consistent with impaired “post-immune” tolerance associated with FcγRIIB dysfunction, though this difference was not seen in the CD19⁺IgD⁻CD27⁺CD38⁻ population. These human data suggest that, even in a healthy heterogeneous population, reduced FcγRIIB function impacts upon post-immune, rather than central/pre-immune tolerance, consistent with the differential impact of FcγRIIB on pre- and post-immune tolerance seen in the mouse, but with notable species-specific differences.”

Moreover this is also discussed in the Discussion section (p16):

“This seems also to be the case in humans, as complementary findings were made in healthy volunteers bearing a loss-of-function variant in *FCGR2B*. The proportion of IGHV4-34 clones, known to be autoreactive, was significantly higher in the activated CD19⁺IgD⁻CD27⁻CD38^{mid/hi} and CD19⁺IgD⁺CD27⁺ B cell subsets in individuals homozygous for the *FCGR2B* T232 loss-of-function SLE-associated polymorphism. This is consistent with a relative impairment of post-immune tolerance in people with reduced FcγRIIB function, as was seen in mice. Mouse and human results were not fully concordant (there was, for example, no evidence in humans of increased pre-immune tolerance associated with reduced FcγRIIB function, at least as this is reflected in IGHV4-34 clone frequency). Nonetheless, human data support a predominant role for FcγRIIB in post-immune tolerance, as was seen in the various mouse models. “

Reviewer #3 (Germinal center response, autoimmune)(Remarks to the Author):

The authors demonstrate interesting opposing effects of FcγRIIB on B cell selection. While FcγRIIB has a well-known inhibitory role during antigen-dependent B cell activation and in that way inhibits inappropriate B cell activation, during central tolerance induction it seems to have the opposite effect.

This shows during development in fewer autoreactive HEL-specific self-reactive B cells being selected, if membrane expressed HEL autoantigen is present and FcγRIIB is absent, and more self-reactive B cells if FcγRIIB is overexpressed. The fewer B cells developing in absence of FcγRIIB are anergic. During antigen-dependent activation, however, in the absence of FcγRIIB, more self-reactive B cells seem to enter GC responses. Similar results are found when weaker tolerance induction is studied (hosts expressing soluble HEL instead of membrane expressed HEL). It seems that in the absence of FcγRIIB non-antigen-specific autoreactive B cells are allowed to enter GC responses. V-gene repertoire expression data from humans support this.

It is concluded that FcγRIIB has opposing roles during preimmune repertoire selection and during antigen-induced B cell differentiation. This makes sense: In the preimmune phase FcγRIIB inhibits BCR signalling, therefore counteracts tolerance induction induced by inappropriate BCR signals. In the antigen-dependent phase of B cell differentiation, FcγRIIB also inhibits BCR signalling. Now, paradoxically, FcγRIIB leads to better control of autoreactive B cells.

The authors make the point that FcγRIIB inhibits recruitment of non-specific “bystander” cells into the GC response. I am having problems with that definition. It is difficult to measure whether GC B cells are really non-specific or just very low affinity. Therefore, there is no way of knowing whether what is seen is real non-specific bystander activation or just a broader activation of cross-reactive B cells.

We agree that it is impossible to fully exclude “bystander” cells having very low affinity for the immunising antigen (affinity which may even be below the detection threshold, though whether such interactions would be of physiological importance is unknown). To minimize this possibility we took advantage of the SW_{HEL} system. Mice were immunized with SRBC only (Figure 6) and we observed that transgenic HEL-specific GC B cells were increased in absence of FcγRIIb in recipient mice in which HEL was an autoantigen. Although we cannot exclude that these cells might be polyreactive and also recognize a determinant of SRBC, this does not seem the most likely explanation. We agree with the Reviewer, however, that one can never exclude such low affinity interactions, so we need to acknowledge that such cells may be contained within a “bystander” population. We have therefore explained this in the Discussion (p 14), while also adding references to recent papers supporting the contribution of non-cognate B cells to on-going GCs (Silver *et al.* J Exp Med 2018 and Degn *et al.* Cell 2017):

“...FcγRIIb controlled the expansion of “bystander” B cells in the GC. Such “bystander” cells, with no measurable affinity for the immunizing antigen, have recently been implicated in contributing to on-going GCs (Silver *et al.* J Exp Med 2018 and Degn *et al.* Cell 2017), supporting our observation. It should be noted, however, that while low affinity cross-reactivity between the transgenic anti-HEL BCR and SRBC antigens cannot be measured, and would seem unlikely, the presence in the “bystander” population of cells with very low affinity for a given antigen cannot be excluded, and could contribute to subsequent measurable “bystander” responses. Nonetheless, no or low....”.

The data presented on antigen-dependent activate are somewhat incomplete: In Fig. 4 and 5 cells entering GC responses are only given as percentages (HEL-specific cells per GC cells), but not in absolute numbers of HEL-specific cells and no size of the GC response is shown. Therefore, it is not clear whether the number of antigen-specific cells recruited into GCs is high or low and how absolute numbers compare between groups. Further, the upper half of Fig. 4b shows only 3 plots each. Are these from wt or mHEL hosts? And how do wt or mHEL hosts compare? Also, Fig. 4e is problematic. Dividing percentages does not make much sense, if the size of the parent populations is not known. Dividing absolute numbers would be better.

In Fig. 6, however, we are told absolute numbers of antigen-specific cells without being shown relative percentages. Therefore, again the reader cannot conclude whether the interpretation of the authors - that fewer non-specific B cells enter GCs in the absence of FcRIIb - is true or whether just total GC sizes are changing.

We are grateful for the reviewer’s comments. We have now added absolute numbers for figures 4 and 5 and Supplementary figure 6. This greatly strengthens the results and confirms the differences observed with frequencies. Moreover, this allows a better visualization of FcγRIIb influence on each subset. The text has been changed accordingly.

The upper half of figure 4b shows representative gating from mHEL recipients. We are presenting below the gating for WT host for the reviewer benefit. As shown, the percentage of total GC B cells in WT and mHEL recipients are roughly similar, with in both cases enhanced GC formation in absence of FcγRIIb compared to WT and BTG groups:

Minor point:

Fig. 4F: what does the axis label mean (5.105 cells)?

We are sorry for this error that has now been corrected. It should have read “HEL⁺ AFCs / 5x10⁶ cells”.

Reviewers' comments:

Reviewer #1 (Remarks to the Author):

The revised version of the manuscript is greatly enhanced compared to the first submitted version, and the response to this Reviewer's questions and requests have been satisfactorily answered.

Reviewer #2 (Remarks to the Author):

The description of human B cell subsets is now much improved however there are still some areas that are not clear.

Figure S7 is an important one in which the B cell subsets are defined. There are problems with the images that may be a pdf conversion problem. The legend is also missing.

The criteria for designating positivity for CD38, or mid/ hi should be made clearer. The cut off between CD38+ and CD38- is between 100 and 1000 on the CD38 axis for the CD27+IgD- cells. In contrast this is from 10 and above for the CD27-IgD- cells designated mid/ hi . Why is this? To me 'mid' implies a low level of positivity. Are CD27+ subsets designated CD38+ actually mid/ negative?

There is an illustration of CD38 and CD24 expression for the CD27-IgD+ cells. It's not clear to me why this is needed. Perhaps this can be explained in the legend or omitted.

Reviewer #3 (Remarks to the Author):

Thank you for the changes. I am happy about how my comments have been addressed.

Point-by-point response to reviewers' comments:

We would like to thank the three reviewers for their constructive comments that have helped improving greatly our manuscript. Our responses to reviewer 2 comments are below.

Reviewer #1 (Remarks to the Author):

The revised version of the manuscript is greatly enhanced compared to the first submitted version, and the response to this Reviewer's questions and requests have been satisfactorily answered.

Reviewer #2 (Remarks to the Author):

The description of human B cell subsets is now much improved however there are still some areas that are not clear.

Figure S7 is an important one in which the B cell subsets are defined. There are problems with the images that may be a pdf conversion problem. The legend is also missing.

The criteria for designating positivity for CD38, or mid/ hi should be made clearer. The cut off between CD38+ and CD38- is between 100 and 1000 on the CD38 axis for the CD27+IgD- cells. In contrast this is from 10 and above for the CD27-IgD- cells designated mid/ hi . Why is this? To me 'mid' implies a low level of positivity. Are CD27+ subsets designated CD38+ actually mid/ negative?

There is an illustration of CD38 and CD24 expression for the CD27-IgD+ cells. It's not clear to me why this is needed. Perhaps this can be explained in the legend or omitted.

The legend has now been added for figure S7.

We agree that the CD38CD24 panel is not required and has now been removed.

We apologize for the quality of the figure that has now been improved.

For the CD38 gating, we defined positivity based on an unstained sample. On this basis, a MFI over 10 was considered positive. The IgD-CD27+ memory B cells express low/intermediate levels of CD38 (MFI from 1 to 250) with a large proportion of the cells expressing intermediate levels (MFI between 10 and 250) while IgD-CD27+ plasmablasts express very high levels of CD38 (MFI over 250). For the IgD-CD27- B cells we focused on activated cells expressing CD38 from intermediate (MFI from 10 to 250) to high (MFI over 250) level. Although most of the cells in this subset expressed intermediate level of CD38, a fraction expressed high level of this marker. We apologise that this was not clear, and we have now clarified this in the text by redefining CD27+IgD- to the more accurate label of CD19+/CD27+IgD-/CD38low/mid. All text, figures, legends and tables have been updated accordingly to be consistent throughout.

Reviewer #3 (Remarks to the Author):

Thank you for the changes. I am happy about how my comments have been addressed.